# NEURO-SYMBOLIC RULE LISTS

## ABSTRACT

Machine learning models deployed in sensitive areas such as healthcare must be interpretable to ensure accountability and fairness. Rule lists (**if** Age $<$ 35 $\wedge$ Priors $> 0$ **then** Recidivism = True, **else if** Next Condition . . . ) offer full transparency, making them well-suited for high-stakes decisions. However, learning such rule lists presents significant challenges. Existing methods based on combinatorial optimization require feature pre-discretization and impose restrictions on rule size. Neuro-symbolic methods use more scalable continuous optimization yet place similar pre-discretization constraints and suffer from unstable optimization. To address the existing limitations, we introduce NYRULES, an end-to-end trainable model that unifies discretization, rule learning, and rule order into a single differentiable framework. We formulate a continuous relaxation of the rule list learning problem that converges to a strict rule list through temperature annealing. NYRULES learns both the discretizations of individual features, as well as their combination into conjunctive rules without any pre-processing or restrictions. Extensive experiments demonstrate that NYRULES consistently outperforms both combinatorial and neuro-symbolic methods, effectively learning simple and complex rules, as well as their order, across a wide range of datasets.

## 1 INTRODUCTION

Machine learning models are increasingly used in high-stakes applications such as healthcare (Deo, 2015), credit risk evaluation (Bhatore et al., 2020), and criminal justice (Lakkaraju & Rudin, 2017), where it is vital that each decision is fair and reasonable. Proxy measures such as Shapley values can give the illusion of interpretability, but are highly problematic as they can not faithfully represent a non-additive models decision process (Gosiewska & Biecek, 2019). Instead, Rudin (2019) argues that it is crucial to use inherently interpretable models, to create systems with human supervision in the loop (Kleinberg et al., 2018).

For particularly sensitive domains such as stroke prediction or recidivism, so called *Rule Lists* are a popular choice (Letham et al., 2015) due to their fully transparent decision making. A rule list predicts based on nested "if-then-else" statements and naturally aligns with the human-decision making process. Each rule is active if its conditions are met, e.g. "**if** Thalassemia = normal $\wedge$ Resting bps $< 151$", and carries a respective prediction, i.e. "**then** $P(\text{Disease}) = 10\%$". A rule list goes through a set of rules in a fixed order, and makes its prediction using the first applicable rule. We show an example rule list for the Heart disease (Detrano et al., 1989) learned with our method in Figure 1, which is highly accurate and easy to understand.

Inferring rule lists from data is a challenging combinatorial optimization problem, as there are super exponentially many options in the number of features. Existing approaches use greedy heuristics (Proenca & van Leeuwen, 2020), sample based on fixed priors (Yang et al., 2017) and even attempt to find globally optimal solutions (Angelino et al., 2018) using branch-bound. However, they all depend on the pre-discretization of continuous features, which in practice deteriorates their performance. That is, features such as "Resting bps" are typically discretized using $2 - 5$ fixed thresholds. Increasing the granularity of pre-processing leads to a combinatorial explosion of the search space, which creates issues in scalability for exact methods and in accuracy for heuristics.

Recently, following the paradigm of neuro-symbolic learning, methods based on continuous optimization were proposed to solve rule learning problems with gradient descent (Wang et al., 2021; Qiao et al., 2021). Nevertheless, neural methods too require to pre-discretize of the features. Most

**if** Chest pain = asymptomatic $\wedge$ Exercise chest pain = 0 $\wedge$ 0.88 < ST depression < 5.24
  **then** P(Disease) = 94%
**else if** Chest pain = asymptomatic $\wedge$ 45.50 < age < 66.42 $\wedge$ Sex = female
  **then** P(Disease) = 87%
**else if** Resting bps < 151 $\wedge$ Ex. chest pain = 0 $\wedge$ ST depr. < 2.65 $\wedge$ Thalassemia = normal
  **then** P(Disease) = 10%
**else if** Chest pain = not ( atypical $\vee$ asymptomatic) $\wedge$ Resting bps < 176 $\wedge$ Max HR > 137
  **then** P(Disease) = 15%
**else if** Chest pain = asymptomatic $\wedge$ 1.46 < ST depression < 5.00
  **then** P(Disease) = 58%
**else** P(Disease) = 52%

Figure 1: Rule list learned with NYRULES on the Heart Disease dataset. NYRULES jointly optimizes thresholds, rule aggregation and ordering into a rule list.

related is the work by Dierckx et al. (2023), who extend neuro-symbolic learning to rule lists by a fixed ordering layer such that the neural network resembles a proper rule list. However, their inability to adapt rule order or thresholds results in unstable training and often subpar accuracy.

To overcome all limitations of prior works, we propose NYRULES, a novel method for learning rule lists differentiably and end-to-end. That is, we unify the learning of predicates, their assembly into rules, and the final order of the rule list into a single architecture. Instead of relying on pre-discretized features, NYRULES learns the discretization of the features as well as which features to aggregate to conjunctive rules. We employ soft approximations of threshold functions (predicates) and combine them using a novel differentiable logical conjunction, which alleviates vanishing gradients issues of previous work. Finally, we introduce a learnable rule priority that is grounded into a strict ordering at the end of training. Altogether, this gives us a holistic differentiable relaxation of rule lists, which can be learned end-to-end. Empirically, we show that NYRULES outperforms the state-of-the-art on a plethora of datasets, especially where exact thresholding is required.

## 2 PRELIMINARIES

We consider a dataset of $n$ pairs $\{(\mathbf{x}, y)\}_{i=1}^{n}$ consisting of the *descriptive feature vector* $\mathbf{x} \in \mathcal{X}$ with $d$ real-valued features $x_i \in \mathbb{R}$, and the discrete-valued *target label* $y \in \mathcal{Y}$. We assume each sample $(\mathbf{x}, y)$ to be a realization of a pair of random variables $(X, Y) \sim P_{X,Y}$, drawn iid.

### 2.1 RULES

We consider predictive rules $r : \mathcal{X} \to \mathcal{Y}$ for supervised classification, which map input samples to predictions. A rule is comprised of an *antecedent* $a : \mathcal{X} \to \{0, 1\}$, which determines whether the rule is applicable to a sample $\mathbf{x}$ or not. Should the antecedent's condition be met, the *consequent* $c \in \mathcal{Y}$ governs the models prediction. A predictive rule is defined as

$$ r(x) = \left\{ \begin{array}{ll} c & \text{if } a(x) = 1 \\ \bar{c} & \text{else} \end{array} \right. $$

If the antecedent is not met, an alternate prediction $\bar{c} \in \mathcal{Y}$ is made. In rule lists, which are introduced shortly, the *else* case is covered by yet another rule.

The antecedent of a rule must be interpretable. In particular, we examine rules given in form of a *logical conjunction* of predicates $\pi$, e.g. the rule "**if** Thalassemia = normal $\wedge$ Resting bps < 151" from the introduction. Each predicate $\pi_i$ represents the presence/absence of a certain characteristic in an individual $\mathbf{x}$, e.g "Resting bps < 151". For tabular data, where $\mathcal{X} = \mathbb{R}^d$, clear and meaningful semantic concepts are usually defined as *parameterized thresholding* functions on individual feature variable $X_i$, i.e.

$$ \pi(x_i; \alpha, \beta) = \mathbb{1}\left[\alpha \le x_i \le \beta\right] \ . $$

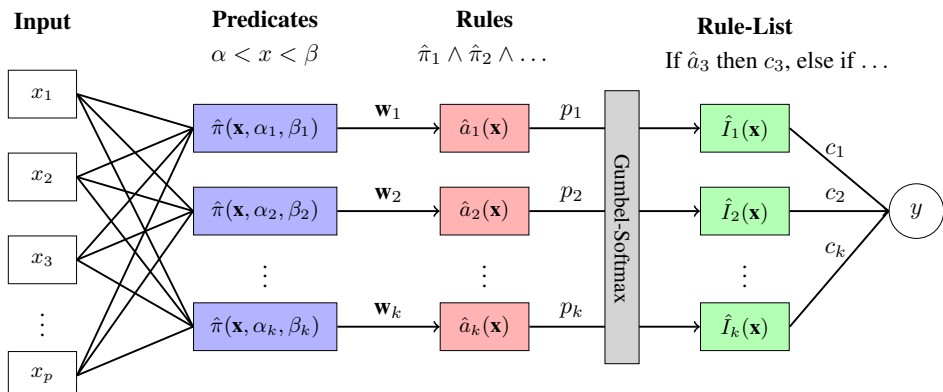

Figure 2: NYRULES architecture. The input $\mathbf{x} \in \mathbb{R}^d$ is discretized into soft predicates $\hat{\pi}$ using learnable threshold $\alpha_j, \beta_j \in \mathbb{R}^d$ and then combined into $k$ rules $\hat{a}_j(\mathbf{x})$. The rules are sorted by their priority $p_j$, where using the Gumbel-Softmax function, we approximate the indicator function $I_j$ of the active rule with highest priority $\hat{I}_j(\mathbf{x})$. The final prediction is computed by taking the weighted sum of the consequents $c_j$ and indicator $I_j$.

As we are considering only rules based on logical conjunctions, the conjunction of multiple predicates on the same feature can always be represented by a single set of thresholds. If no condition is placed on a feature $x_i$, then the corresponding thresholds are given by $-\infty$ and $\infty$ resp. This allows us to define the class of antecedent $a$ rule functions as a logical conjunction of predicates as per

$$a(\mathbf{x}; \theta) = \bigwedge_{i=1}^{d} \pi\left(x_i; \alpha_i, \beta_i\right) \ ,$$

parameterized by $\theta \in \mathbb{R}^{2*d}$. Hence, rule-based methods aggregate multiple rules into a performant classifier that is easily human interpretable.

**Rule lists** (Cohen, 1995) model nested if-then-else classifiers. To make a prediction, we traverse the set of rules in a fixed order and use the consequent of the first rule, whose antecedent applies. Given a set of $k$ rules $(a_j, c_j)$, each rule is assigned its *unique* priority $p_j \in \mathbb{N}$. To classify a sample $\mathbf{x}$ with a rule list $rl : \mathcal{X} \to \mathcal{Y}$, that rule $r_j$ with the highest priority is used whose antecedent holds, i.e.

$$rl(\mathbf{x}; \Theta, \mathbf{p}) = c_j$$
$$\text{such that } a_j(\mathbf{x}; \theta_j) = 1 \ \wedge \ (\forall i \neq j : a_i(\mathbf{x}) = 0 \vee p_j > p_i) \ . \tag{1}$$

Informally, a rule list is as a nested if-then-else structure, as commonly used in programming. The *else* case is realized through an always-on rule with the lowest priority.

## 3 DIFFERENTIABLE RULE LISTS

In this section we introduce **Neu**ro-Symbolic **Rule** Lists, or short NYRULES. We show the architecture of NYRULES in Figure 2. The first step is to transform the continuous input features into binary **predicates** such as "$18 <$Age$< 30$" to use as a basic building blocks for rule construction. In contrast to all existing methods, NYRULES avoids the need for pre-discretization and instead learns a task specific thresholding of the features.

Next, we combine the learned predicates into **rules** $\hat{a}_j(\mathbf{x})$, where $\hat{a}$ is a differentiable function that behaves like a logical conjunction for binary predicates, but is also able to handle soft predicates $\hat{\pi} \in [0, 1]$. In particular, our formulation encourages interpretable, succinct rules using weights $\mathbf{w}_j$ and alleviates the problem of vanishing gradients compared to previous formulations.

Finally, we aggregate a set of $k$ rules into a **rule list** rule list as introduced in Eq. (1). $I_j(\mathbf{x})$ indicates if rule $j$ is to be used for the prediction, i.e. whether it is active ($\hat{a}(\mathbf{x}) = 1$) and has the highest

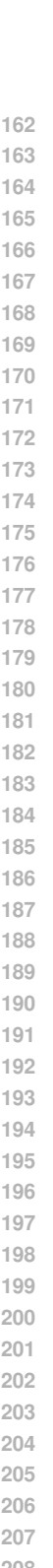
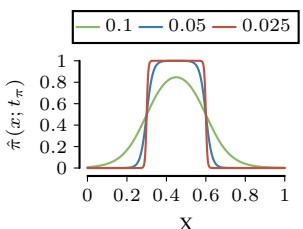
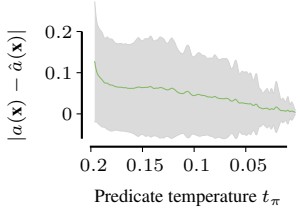
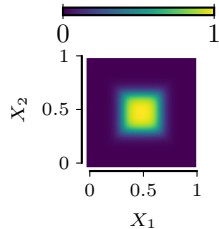

(a) Soft predicate $\hat{\pi}(x; t_\pi)$ for different temperatures $t_\pi$.

(b) Softness of rule $\hat{a}(x)$ for different temperatures $t_\pi$. The grey corresponds to the variance.

(c) Differentiable logical conjunction $\hat{a}(\mathbf{x})$ over two features.

Figure 3: The soft predicate with different temperatures (a) approaches the true thresholding with decreasing temperature (b). Multiple soft predicates are combined into a conjunctive rule (c).

priority $p_j$. We use the **Gumbel-Softmax** function Jang et al. (2017a) to approximate this indicator and formulate a differentiable relaxation for the entire rule list that allows us to jointly optimize a rule list from discretization parameters to rule order. All our approximations converge towards strict logical operators at the end of training, i.e. NYRULES converges to a crisp rule list.

## 3.1 CONJUNCTIVE RULES

We begin with the antecedent $a$ of a rule that makes up the "**if** ..." condition of a rule.

**Thresholding Layer.** As the building blocks of every rule we consider thresholding functions $\pi(x_i; \alpha, \beta) = \mathbb{1}\left[\alpha \le x_i \le \beta\right]$ on individual features $x_i$ as predicates $\pi$. It is common practice to employ equal-width/-frequency binning for continuous features $X_i$ and thus obtain a fixed number of predicates. The potential solution may become more accurate by increasing the number of bins, but also comes at a much higher optimization difficulty.

Neural rule learning methods also require to pre-discretize continuous features. since the thresholding function is not continuous at the bounds $\alpha$ and $\beta$, and has gradient of $0$ elsewhere. To address these issues, we use the soft binning function as introduced by Yang et al. (2018) as

$$\hat{\pi}(x_i; \alpha, \beta, t_\pi) = \frac{e^{\frac{1}{t_\pi}(2x_i - \alpha_i)}}{e^{\frac{1}{t_\pi}x_i} + e^{\frac{1}{t_\pi}(2x_i - \alpha_i)} + e^{\frac{1}{t_\pi}(3x_i - \alpha_i - \beta_i)}} \ .$$

The resulting function approximates a thresholding function, where the softness of said function is controlled by a temperature parameter $t_\pi$. We show $\hat{\pi}$ using different temperatures $t_\pi$ in Figure 3a. Just as the Figure suggests, in the limit $t_\pi \to 0$ the soft predicate converges to the true thresholding function $\pi(x_i; \alpha, \beta)$ (shown in Appendix A.1), i.e. $\lim_{t_\pi \to 0} \hat{\pi}(x_i; \alpha, \beta, t_\pi) = \pi(x_i; \alpha, \beta)$ . Therefore, we use temperature annealing to increase crispness of the predicates $\alpha$ and $\beta$ as the training progresses. We begin start with a higher temperature $t_\pi$, which results in smoother predicates $\hat{\pi}$ and avoids exploding/vanishing gradients with respect to $\alpha, \beta$. That means that initially, our predicates are not strictly binary but soft instead, i.e. $\hat{\pi}(x_i) \in [0, 1]$.

In the end, we seek to obtain strict logical rules for use in a rule list. Hence, we continuously decrease the temperature $t_\pi$ so that in the end $\forall x_i : \hat{\pi}(x_i) \approx 1 \lor \hat{\pi}(x_i) \approx 0$, except at the boundaries itself, where if $x_i = \alpha \lor x_i = \beta : \hat{\pi}(x_i) = \frac{1}{2}$. We show the softness of rules based on soft predicates in Figure 3a. When the temperature is annealed close to zero, the predicates become increasingly binary and the difference to the true thresholding function vanishes.

**Logical conjunction.** To learn rule antecedents, we need to flexibly combine the trainable predicates $\hat{\pi}(x_i)$ into a logical conjunction. To this end, we introduce a differentiable logical conjunction function $\hat{a}$ specifically designed for soft predicates. We omit from the notation $\alpha, \beta$ and $t_\pi$ for brevity. We require the logical conjunction to satisfy three conditions:

$$\hat{a}(\pi) = \begin{cases} 1, \text{if } \forall \pi(x_i) = 1 \\ 0, \text{if } \exists \pi(x_i) = 0 \\ \in [0, 1] \text{ else.} \end{cases}$$

We base our approach on the *reciprocal* of the predicate $\hat{\pi}(x)^{-1}$, aggregated by the weighted harmonic mean proposed by Xu et al. (2024), which is given by

$$\hat{a}(\mathbf{x}) = \frac{\sum_{i=1}^{d} w_i}{\sum_{i=1}^{d} w_i \hat{\pi}(x_i)^{-1}} \ .$$

This function fulfills all the outlined criteria: if $\forall \hat{\pi}(x_i) = 1$, the function evaluates to 1, while if $\exists \hat{\pi}(x_i) = 0 \to \hat{a}(\mathbf{x}) = 0$. Additionally, by setting the corresponding weight of a predicate $w_i$ to 0, the network can disable predicates and thus obtain more succinct rules. Its main drawback is the issue of vanishing gradients in the case of $\hat{\pi}(x_i) = 0$.

The partial derivatives of the rule function $\hat{a}$ with respect to its parameters are given by

$$\frac{\partial \hat{a}(\mathbf{x})}{\partial \hat{\pi}(x_j)} = \frac{w_j \left( \sum_{i=1}^{d} w_i \right)}{\hat{\pi}(x_j)^2 \left( \sum_{i=1}^{d} w_i \hat{\pi}(x_i)^{-1} \right)^2} \ , \quad \frac{\partial \hat{a}(\mathbf{x})}{\partial w_j} = \frac{\sum_{i=1}^{d} w_i (\hat{\pi}(x_i)^{-1} - \hat{\pi}(x_j)^{-1})}{(\sum_{i=1}^{d} w_i \hat{\pi}(x_i)^{-1})^2} \ .$$

Consider the case where there is a predicate $\hat{\pi}(x_l) = 0$ with $w_l > 0$. Then the partial derivative of all other predicates $\hat{\pi}(x_j)$ is zero, as the reciprocal $\hat{\pi}(x_l)^{-1}$ is in the denominator and $\lim_{x_l \to 0} \hat{\pi}(x_l)^{-1} = \infty$. Similar, with respect to the weights $w_i$, the derivative is zero, as the squared reciprocal in the denominator dominates the term. This is a significant issue, as with increasingly crisp predicates $\hat{\pi}(x_i) \to 0$, the gradient vanishes if the rule has an inactive predicate.

We solve this issue by relaxing the requirements of the soft conjunction. That is, we allow a slack of $\epsilon$ when $\exists \hat{\pi}(x_i) = 0$. Hence, we do not require the conjunction to take a value of zero but only $\hat{a}(\mathbf{x}) \le \epsilon$ instead. To this end, we modify the reciprocal and resulting logical conjunction using a weight dependent constant $\eta$ as

$$\eta = \frac{\epsilon}{\sum_{i=1}^{d} w_i} \ , \quad \hat{a}(\mathbf{x}) = \frac{\sum_{i=1}^{d} w_i}{\sum_{i=1}^{d} w_i \frac{1+\eta}{\hat{\pi}(x_i)+\eta}} \ .$$

With this relaxed formulation, we now obtain gradients that do not vanish once a predicate is inactive. We show in Appendix A.3 that in the limit for all inactive predicates $\forall i : \hat{\pi}(x_i) = 0$ the partial derivatives with respect to the predicates $\frac{\partial \hat{a}(\mathbf{x})}{\partial \hat{\pi}(x_j)} \approx \frac{w_j}{\sum_{i=1}^{d} w_i}$ are still non-zero, whereas the $\frac{\partial \hat{a}(\mathbf{x})}{\partial w_j} > 0$ if there is at least one active predicate $\hat{\pi}(x_i) > 0$ with $w_i > 0$. That is, the gradient does not vanish anymore if a rule contains an inactive predicate. Intuitively, the weight-dependent constant $\eta$ automatically adjusts the amount of slack such that when the rule is mostly inactive, i.e. most $w_i$ are small, the slack is increased and the gradient flow in this rule increases. While if a rule is mostly active, the slack is decreased and the gradient is not influenced by $\eta$. We perform an ablation in Section 5.2, where we observe that using the relaxed conjunction instead of its unbounded counterpart $\hat{a}(\mathbf{x}) \le \epsilon$ improves the average $F_1$ score by $\mathbf{1.7}$x and in some cases even by $4$x.

**Differentiable Rule.** With the predicates and the logical conjunction in place, the rule antecedent

$$\lim_{t_\pi \to 0} \hat{a}(\mathbf{x}) = \begin{cases} 1 & \text{if } \forall i : w_i = 0 \lor \hat{\pi}(x_i) = 1 \\ 0 & \text{else} \end{cases}$$

can be flexibly learned. As the rule consequent, which is the "then ..." part of a rule, we use a logit vector $\mathbf{c} \in \mathbb{R}^l$ in the classification setting with $l$ classes. A singular rule is then defined as

$$\text{if } \hat{a}(\mathbf{x}) = 1 \text{ then } \arg \max_{m \in \{1, \dots, l\}} c_m \ , \text{ else } \dots \ .$$

The "**else**" case in a rule list is handled by a subsequent rule, the mechanism of which we will discuss in the following section.

## 3.2 SOFT RULE LISTS

A rule list consists of a set of $k$ rule tuples $\{(a_j, c_j, p_j)\}_{j=1}^{k}$, made up of an antecedent $a_j : \mathcal{X} \to \{0, 1\}$, a consequent $c_j \in \mathbb{R}^l$ and a unique priority $p_j \in \mathbb{R}^+$. A sample is classified using the

prediction of the rule with highest priority and active antecedent as per Eq. (1). To allow for continuous optimization, we reformulate the $rl(\mathbf{x})$ as a linear combination of consequents $c_j$. That is, we combine the antecedent $a_j$ and priority $p_j$ into the *active priority* $a_j^p$ as

$$a_j^p(\mathbf{x}) = a_j(\mathbf{x}) \cdot p_j \ .$$

The arg max indicator $I_j(\mathbf{x}) = \mathbb{1}[j = \arg\max_{j \in \{1,\dots,k\}} a_j^p(\mathbf{x})]$ of the active priority vector $\mathbf{a}^p$ indicates with which rule the prediction is to be made. Hence, we can re-write the rule list simply as $rl(\mathbf{x}) = \sum_{j=1}^k I_j(\mathbf{x})c_j$.

**Continuous Relaxation.** To learn a rule list, we initially relax the constraint to $\sum_{j=1}^k \hat{I}_j(\mathbf{x}) = 1$ and hence allow multiple rules to contribute, weighted by their active priority $a_j^p$. To this end, we use the *Gumbel-Softmax* (Jang et al., 2017b), which is a variant of the reparameterization trick and provides a differentiable approximation to the argmax function. Given the active priority $\mathbf{a}^p$ and a temperature $t_{rl}$, the Gumbel-Softmax is defined as

$$\hat{I}_j(\mathbf{x}; t_{rl}) = \text{softmax}\left(\frac{\mathbf{a}^p + \mathbf{g}}{t_{rl}}\right) \ ,$$

where $\mathbf{g} \in \mathbb{R}^k$ is random noise sampled from the Gumbel distribution. The Gumbel-Softmax approach interpolates between the strict one-hot encoding of a rule list and a linear combination weighted by the rule priorities. In particular, in the limit $t_{rl} \to 0$, the Gumbel-Softmax converges to the arg max function. Thus, the soft rule list is given by

$$\widehat{rl}(\mathbf{x}) = \sum_{j=1}^k c_j \cdot \hat{I}_j(\mathbf{x}) \ .$$

We plot the impact of the relaxation with regard to different temperatures $t_\pi$ in Figure 4. We start training with a positive temperature $t_{rl} = 0.5$, where the rule with highest active priority has on average $0.75$ of the weight, whilst the other rules contribute the remaining $0.25$. We continuously decrease the temperature $t_{rl}$ towards zero, so that in the end the indicator $\hat{I}_j$ of the actual rule dominates with a weight of $0.99$. With an appropriate annealing schedule NYRULES starts training using a relaxed rule list, which it can optimize, and continuously moves towards a strict rule list. After

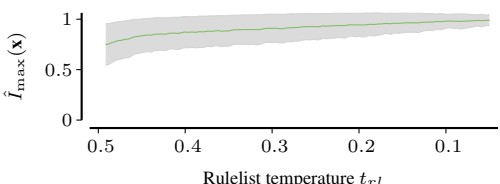

Figure 4: Weight of highest priority rule $\hat{I}_{\max}(\mathbf{x})$ during training with decreasing temperature $t_{rl}$. The grey corresponds to the variance.

training, we convert the soft rule list into a crisp rule list. We construct each rule $r(\mathbf{x})$ as a conjunction of all predicates with $a_i > 0$ and use as lower/upper bound the parameters $\alpha_i$ and $\beta_i$. We then order the rules based on their priority $p_j$ and construct a strict rule list $rl(\mathbf{x})$. To learn the antecedents, we parameterize $c_j = \text{softmax}(\hat{c}_j)$, where $\hat{c}_j \in \mathbb{R}^l$ is a learnable vector i.e. the logits.

### 3.3 ENTIRE ARCHITECTURE

Our model takes as input any real-valued feature vector $\mathbf{x} \in \mathbb{R}^d$, where we first one-hot encode the categorical features into binary features. For any instantiation of our model, the number of rules $k$ and the number of classes $l$ is fixed beforehand. We show the resulting architecture in Figure 2.

First, NYRULES discretizes the input features into $j \in \{1,\dots,k\}$ sets of $d$ predicates such as "18 $<$Age $<$ 30" or "Diabetes = True". Each set of predicates is then composed into the antecedent of rule $j$, using the respective weights $\mathbf{w}_j \in \mathbb{R}^d$. The activation vector $\hat{\mathbf{a}} \in [0,1]^k$ represents the activation of rules such as "**if** Age $<$ 30 $\wedge$ Diabetes = True".

Next, NYRULES computes the active priority $a_j^p = \mathbf{r} \circ \mathbf{p}$ and passes it through the Gumbel-Softmax function to obtain the approximate arg max of the rule list $\hat{\mathbf{I}}$. The prediction $rl(\mathbf{x})$ is computed by taking the weighted sum of consequents $rl(\mathbf{x}) = \sum_{j=1}^k \hat{I}_j \cdot c_j$.

### 3.4 OBJECTIVE

Lastly, we turn to the learning setup. We train NYRULES in a standard supervised learning setting given an arbitrary loss function $\ell$ and a sample of the data distribution $P_{X,Y}$. In the following, we use the cross-entropy loss for binary classification tasks, i.e. $\ell(rl(\mathbf{x};\Theta),y) = -y\log(rl(\mathbf{x};\Theta)) - (1-y)\log(1 - rl(\mathbf{x};\Theta))$, though in principle any differentiable loss function $\ell$ can be used.

**Minimum-Support** Besides the chosen loss function, we propose a regularization term to ensure that each rule represents a non-trivial amount of points. That is, akin to the minimum support requirement in classical rule lists or decision trees, we penalize rules that are never used or used too often. To this end, we add a regularization term based off the rule usage indicator $\hat{I}_j(\mathbf{x}_i)$. We compute the coverage, i.e. the fraction of points where each individual rules is active, over the training set $\{x_i\}_{i=1}^n$ as $cov_j = \frac{1}{n}\sum_{i=1}^n \hat{I}_j(\mathbf{x}_i)$. The support regularizer is then given by

$$\mathcal{R}(\Theta) = \frac{1}{k}\sum_{j=1}^k \max\left(0, cov_{\min} - cov_j\right)^2 + \max\left(0, cov_j - cov_{\max}\right)^2 \ ,$$

where $cov_{\min}$ and $cov_{\max}$ are user-specified minimum and maximum supports. We weight the regularization term using a hyperparameter $\lambda$. The overall objective given a training set $\{(\mathbf{x}_i, y_i)\}_{i=1}^n$ is then to optimize the rule list parameters $\Theta = (\beta_1, \alpha_1, \ldots, \beta_k, \alpha_k, \mathbf{w}_1, \ldots, \mathbf{w}_k, \mathbf{p})$ as

$$\arg\min_{\Theta} \frac{1}{n}\sum_{i=1}^n \ell(rl(\mathbf{x}_i;\Theta), y_i) + \lambda\mathcal{R}(\Theta) \ .$$

## 4 RELATED WORK

Rule lists were introduced in the early 90s (Cohen, 1995) and have since been used in various applications, such as healthcare (Deo, 2015), criminal justice (Angelino et al., 2018), and credit risk evaluation (Bhatore et al., 2020). Similarly, decision trees (Breiman, 2017), which can also be easily transformed into rules by tracing the path from the root to the leaf, have also been widely used in practice. The approaches use greedy combinatorial optimization to find the best rule set. Instead of relying on the greedy growing of the model, Wang et al. (2017); Yang et al. (2017) propose Bayesian rule lists, a probabilistic model, where the complexity of the model is controlled by a prior, which is specified by the user. In practice, these priors result in short rule lists but can harm the performance if misspecified. To automate the trade-off between complexity and accuracy, Proenca & van Leeuwen (2020); Papagianni & van Leeuwen (2023) propose an MDL-based approach, which uses an MDL-score for model selection. In practice, this results in more accurate and concise rule lists compared to previous approaches. There are also approaches that attempt to find optimal rule lists (Angelino et al., 2018; Aivodji et al., 2022). Due to the expensive combinatorial optimization, exact methods are not applicable beyond trivially sized datasets and have to severely restrict the search space in terms of rule size, feature quantization and number of rules.

Instead of using combinatorial optimization, neuro-symbolic approaches have been proposed to learn rule classifiers. Qiao et al. (2021) proposes the first approach to learn rule sets in an end-to-end scheme. They formulate a novel neural architecture that uses continuous relaxations of logical operators. Here, techniques from the field of fuzzy logics are used to differentiably optimize logical operations such as negation, disjunction and conjunction (van Krieken et al., 2022). After training, the rules are extracted from the network. This is extended by Wang et al. (2021) to a deeper architecture that allows to learn more complicated rules; however, this often results in worse interpretability. Walter et al. (2024) propose to learn rules for binary data that are not only predictive but also descriptive, which improves explainability but reduces accuracy. Kusters et al. (2022) introduce a differentiable approach to dynamically learn rule predicates, but focuses on linear decision boundaries which can not be translated into interpretable single feature thresholds. Dierckx et al. (2023) extend the approach of Qiao et al. (2021) by introducing a hierarchy among the rules, allowing to learn rule lists. Although these methods resolve the scalability issue, they still rely on pre-discretization of the features, similar to the combinatorial approaches. In contrast, NYRULES learns discretizations on the fly while being highly scalable and accurate.

| | NYRULES | RLNET | RRL | DRNET | GREEDY | CLASSY | CORELS | SBRL | RIPPER | XGBOOST |
|---|---|---|---|---|---|---|---|---|---|---|
| Adult | 0.80 ± 0.01 | 0.76 ± 0.01 | 0.77 ± 0.03 | 0.78 ± 0.01 | 0.75 ± 0.01 | **0.81** ± 0.0 | 0.80 ± 0.0 | 0.67 ± 0.02 | 0.80 ± 0.0 | 0.79 ± 0.01 |
| Android Malware | 0.92 ± 0.0 | **0.95** ± 0.01 | 0.92 ± 0.03 | **0.95** ± 0.01 | 0.86 ± 0.0 | 0.94 ± 0.0 | 0.33 ± 0.01 | n/a | 0.86 ± 0.03 | 0.96 ± 0.0 |
| COMPAS | 0.66 ± 0.01 | 0.65 ± 0.02 | 0.59 ± 0.02 | 0.61 ± 0.02 | 0.66 ± 0.02 | **0.67** ± 0.02 | 0.65 ± 0.01 | 0.32 ± 0.01 | 0.65 ± 0.01 | 0.68 ± 0.01 |
| Covid ICU | 0.62 ± 0.03 | 0.60 ± 0.05 | 0.63 ± 0.03 | 0.48 ± 0.07 | 0.63 ± 0.02 | 0.60 ± 0.06 | 0.61 ± 0.03 | 0.38 ± 0.03 | **0.64** ± 0.02 | **0.64** ± 0.02 |
| Credit Card | **0.79** ± 0.01 | 0.77 ± 0.02 | 0.75 ± 0.05 | 0.75 ± 0.02 | **0.79** ± 0.01 | **0.79** ± 0.01 | **0.79** ± 0.0 | 0.54 ± 0.03 | 0.76 ± 0.05 | 0.68 ± 0.0 |
| German Credit | **0.72** ± 0.03 | 0.71 ± 0.04 | 0.71 ± 0.03 | 0.15 ± 0.02 | 0.67 ± 0.04 | 0.67 ± 0.06 | 0.61 ± 0.04 | 0.58 ± 0.03 | 0.71 ± 0.04 | 0.68 ± 0.02 |
| Credit Screening | **0.86** ± 0.02 | 0.84 ± 0.02 | 0.82 ± 0.03 | 0.43 ± 0.04 | **0.86** ± 0.02 | 0.85 ± 0.02 | 0.74 ± 0.04 | **0.86** ± 0.02 | **0.86** ± 0.02 | 0.84 ± 0.02 |
| Diabetes | 0.73 ± 0.02 | 0.70 ± 0.03 | 0.73 ± 0.07 | 0.44 ± 0.14 | 0.71 ± 0.04 | 0.70 ± 0.04 | 0.70 ± 0.03 | 0.52 ± 0.13 | **0.74** ± 0.07 | 0.71 ± 0.03 |
| Electricity | **0.75** ± 0.0 | 0.69 ± 0.03 | 0.63 ± 0.09 | 0.61 ± 0.01 | **0.75** ± 0.0 | 0.59 ± 0.01 | 0.70 ± 0.01 | 0.47 ± 0.03 | **0.75** ± 0.01 | 0.83 ± 0.0 |
| FICO | **0.70** ± 0.01 | 0.67 ± 0.02 | 0.64 ± 0.03 | 0.63 ± 0.03 | 0.69 ± 0.01 | 0.67 ± 0.02 | 0.63 ± 0.02 | 0.36 ± 0.01 | **0.70** ± 0.01 | 0.71 ± 0.01 |
| Heart Disease | 0.78 ± 0.04 | 0.74 ± 0.02 | 0.72 ± 0.04 | 0.51 ± 0.1 | 0.71 ± 0.05 | 0.77 ± 0.09 | 0.68 ± 0.05 | 0.56 ± 0.21 | **0.80** ± 0.05 | 0.78 ± 0.09 |
| Hepatitis | 0.79 ± 0.06 | 0.77 ± 0.07 | 0.78 ± 0.08 | 0.18 ± 0.04 | 0.73 ± 0.08 | 0.74 ± 0.07 | **0.82** ± 0.04 | 0.70 ± 0.07 | 0.75 ± 0.09 | 0.70 ± 0.07 |
| Juvenile | 0.88 ± 0.02 | 0.87 ± 0.01 | 0.88 ± 0.01 | **0.89** ± 0.01 | 0.83 ± 0.04 | 0.88 ± 0.01 | 0.80 ± 0.02 | n/a | 0.03 ± 0.01 | 0.74 ± 0.03 |
| Magic | **0.79** ± 0.01 | 0.77 ± 0.01 | 0.72 ± 0.03 | 0.77 ± 0.03 | 0.75 ± 0.01 | 0.74 ± 0.01 | 0.72 ± 0.01 | 0.55 ± 0.06 | 0.77 ± 0.0 | 0.85 ± 0.0 |
| Phishing | 0.91 ± 0.01 | 0.93 ± 0.0 | 0.83 ± 0.06 | **0.94** ± 0.0 | 0.89 ± 0.0 | 0.92 ± 0.01 | 0.27 ± 0.01 | 0.87 ± 0.02 | 0.89 ± 0.0 | 0.95 ± 0.0 |
| Phoneme | **0.79** ± 0.02 | 0.71 ± 0.01 | 0.72 ± 0.03 | 0.69 ± 0.02 | 0.76 ± 0.01 | **0.79** ± 0.02 | 0.74 ± 0.01 | 0.71 ± 0.04 | 0.77 ± 0.01 | 0.85 ± 0.01 |
| QSAR | 0.81 ± 0.03 | **0.83** ± 0.02 | 0.80 ± 0.01 | 0.52 ± 0.02 | 0.74 ± 0.03 | 0.82 ± 0.03 | 0.72 ± 0.01 | 0.72 ± 0.02 | 0.79 ± 0.03 | 0.84 ± 0.02 |
| Ring | **0.92** ± 0.02 | 0.81 ± 0.01 | 0.83 ± 0.04 | 0.33 ± 0.02 | 0.56 ± 0.02 | 0.65 ± 0.03 | 0.63 ± 0.02 | 0.68 ± 0.02 | 0.74 ± 0.04 | 0.94 ± 0.0 |
| Titanic | 0.77 ± 0.02 | 0.74 ± 0.03 | 0.69 ± 0.06 | 0.45 ± 0.09 | **0.78** ± 0.02 | 0.77 ± 0.02 | 0.66 ± 0.04 | 0.16 ± 0.02 | 0.75 ± 0.03 | 0.76 ± 0.03 |
| Tokyo | 0.91 ± 0.03 | 0.91 ± 0.02 | 0.91 ± 0.01 | 0.25 ± 0.09 | 0.88 ± 0.01 | **0.92** ± 0.02 | 0.87 ± 0.03 | **0.92** ± 0.01 | **0.92** ± 0.03 | 0.92 ± 0.02 |
| Rank | **2.60** | 4.50 | 5.20 | 6.85 | 4.72 | 4.00 | 5.92 | 7.56 | 3.50 | n/a |

Table 1: Comparison of different rule list methods on 20 real world datasets. Each rule list is set to (maximum) length 10. We report the $F_1$ score under 5-fold cross validation and provide XGBOOST as a benchmark. NYRULES is consistently the best or close to the best performing method.

## 5 EXPERIMENTS

We compare NYRULES against optimal rule lists (CORELS, Angelino et al. 2018), scalable Bayesian rule lists (SBRL, Yang et al. 2017), MDL-based rule lists (CLASSY, Proenca & van Leeuwen 2020), greedily learned rule lists from decision trees (GREEDY), as well the neural rule lists (RLNET, Dierckx et al. 2023). Additionally, we compare with two neuro-symbolic rule set methods, namely RRL (Wang et al. 2021) and DRNET (Qiao et al. 2021), and provide a reference point for achievable performance in form of XGBOOST. We provide the source code in the Supplementary Material.

### 5.1 REAL WORLD

We begin with a comprehensive comparison on 20 real world datasets from domains such as medicine and finance for binary classification. The dataset characteristics and sources can be found in the Appendix B. For all methods, we grid search the best hyperparameter set using 5 hold-out-datasets and use that configuration for all datasets (see Appendix C). To simulate real world conditions, in which interpretability is paramount, we limit each method to a budget of $\{10, 15, \ldots, 30\}$ rules. We report the $F_1$ score weighted by class frequency, averaged over 5-fold cross validation for 10 rules and excluding XGBOOST for rank computation in Table 1, and show the trend for increasing budget in Figure 5a. Experiments that timed out after 24 hours are indicated by $n/a$.

**Overall.** Across all datasets, the neural RLNET and RRL and the heuristic CLASSY and GREEDY are closely matched, achieving an average rank between $3.5 - 4.5$ out of 8 methods, while the exact methods CORELS and SBRL performance is subpar. In contrast, NYRULES performance stands out, achieving an average rank of **2.30**. Across all datasets, NYRULES is either the best or close to the best performing method, which shows the robustness of our method across different domains. The accuracy score, provided in Appendix D, paints a similar picture with NYRULES ranking first.

Amongst all datasets, NYRULES performs particularly well on the `Ring` dataset, where it outperforms the next best method by **0.13** $F_1$ points. Upon closer inspection, the `Ring` dataset uniqueness lies in its exclusively continuous features, making it an ideal benchmark to evaluate the impact of continuous feature handling. Here, the ability of NYRULES to learn exact thresholds becomes a significant competitive edge. For applications in medicine, where continuous biomarkers such as blood pressure or cholesterol levels are often key indicators, NYRULES could potentially improve the accuracy of rule lists whilst maintaining full interpretability. Furthermore, in the Appendix E, we closely investigate under which conditions methods benefit/struggle using synthetic data.

**Rule List Length.** We plot the average $F_1$ score, normalized by the dataset maximum, under an increasing number of rules in Figure 5a. NYRULES remains the best performing method for both short and long rule lists, where for short rule lists GREEDY is closest but deteriorates with more rules, whereas CLASSY is subpar for short rule lists but comes closer with more rules. We further

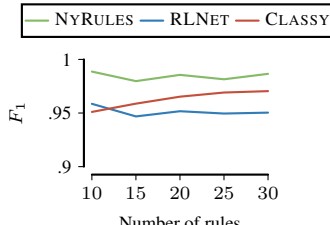 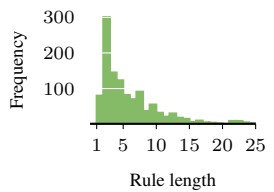 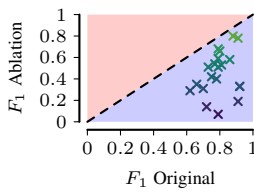

(a) Relative $F_1$-score under increasing number of rules.

(b) Distribution of NYRULES rule lengths across all datasets.

(c) Ablation: $F_1$ score with vs without relaxed conjunction.

Figure 5: NYRULES is accurate for both short and long rule lists **(a)**. The lengths of the learned rules follow a power law **(b)**, and consist of mostly succinct and some detailed rules. Using the relaxed conjunction $\hat{a}(\mathbf{x}) \leq \epsilon$ is always better (blue area) and improves the $F_1$ score on average by **0.3 (c)**.

analyze the length of individual rules learned by NYRULES, i.e. the number of predicates, in Figure 5b, as a proxy measure for the ease of comprehension of the rules learned by NYRULES. In general, shorter rules are easier to understand, though this can comes at a decrease in trust/perceived utility Fürnkranz et al. (2020). The analysis of the distribution of rule lengths across all datasets indicates that they follow a power-law distribution, with a peak at 2 predicates. That is, most rules are simple and some are more complex. Whilst the majority of rules stays below 10 predicates, NYRULES is able to learn rules with up to 25 predicates, which is a testament to its flexibility. In the end, to assess for a particular use case, whether a rule list learned by NyRules is more interpretable and trustworthy than one learned by another method, a user study is required.

**Multi-class Classification.** We focus in this paper on differentiably learning the rules and their order, and less so on the consequents. To allow extensive comparison against all methods, we focus on binary classification. To run NYRULES on multi-class datasets, we simply need to expand the dimension of the consequent vector to the number of classes, i.e. $c \in \mathbb{R}^l$. We provide results for multi-class classification in the Table 2. NYRULES remains the highest ranking method on average, with a rank of $1.50$, and shows that it is not only limited to binary classification.

**Runtime.** Lastly, we examine the scalability of NYRULES in contrast to other rule lists. We provide the average runtime of each method across all benchmarks in Figure 6. NYRULES on average takes 75s per dataset. This is faster than DRNET, RLNET, and SBRL, but significantly slower than the greedy approaches GREEDY, CLASSY and the neural RRL, which all take below 10s per dataset. In general, NYRULES incurs a computational overhead compared to the greedy methods but compensates for it in terms of classification accuracy. RRL optimizes only a rule set instead of a rule list and avoids the more costly rule list optimization, which explains its faster runtime.

## 5.2 ABLATION STUDIES

Finally, we perform an ablation to assess the efficacy of the relaxed conjunction $\hat{a}(\mathbf{x}) \leq \epsilon$. To this end, we re-run NYRULES on all datasets without this adjustment, i.e. with $\epsilon = 0$, and plot the difference to the original $F_1$ score in Figure 5c. We observe that on most datasets the relaxed conjunction outperforms the strict conjunction by a large margin, and on average by 0.3 $F_1$ points. The strict conjunction is not superior on any dataset to the relaxed conjunction. The extent of improvement stresses the importance of non-vanishing gradients and highlights the contribution of relaxing the logical conjunction by NYRULES.

We also perform ablation studies for the thresholding and rule ordering and provide the complete results in the Appendix F. We find the $F_1$ of NYRULES on average is degraded by $0.04$ and $0.03$ resp. for uniform and kmeans based thresholding. The difference is highly dataset and discretization dependent. For example, the performance on `Adult` drops by $0.04$ using uniform but by $0.14$ for kmeans thresholding. In general, while fixed binning can sometimes achieve reasonable results, it requires users to tune the discretization manually. On the other hand, the learned discretization by NyRules performs at least as well as the fixed binning and outperforms it on many datasets.

For the rule ordering, we find that the $F_1$ of NYRULES is degraded by $0.04$ on average, with the largest drop on the `Credit Card` and `Adult` datasets. These datasets contain many samples

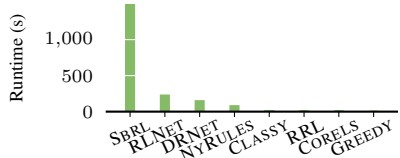

Figure 6: Average runtime over all benchmarked datasets.

| Dataset | NYRULES | RLNET | CLASSY | GREEDY |
|---|---|---|---|---|
| Car | 0.83 | **0.84** | 0.83 | 0.05 |
| ecoli | **0.84** | 0.78 | 0.75 | 0.55 |
| Iris | **0.95** | 0.83 | 0.94 | 0.56 |
| Yeast | **0.54** | 0.41 | 0.52 | 0.21 |
| Avg. Rank | **1.50** | 2.25 | 2.25 | 4.00 |

Table 2: $F_1$ scores for multi-class classification. NYRULES is the best performing method on average.

($n > 30.000$) and likely allow learning of many specific rules with sufficient samples. Here, finding a correct ordering of rules is crucial. Hence, we see that the learned rule ordering can improve the performance of NYRULES on larger datasets. Overall, the ablation studies demonstrate the efficacy of each component of NYRULES, which together allow it to outperform the competition.

## 6 CONCLUSION

We propose NYRULES, a differentiable relaxation of the rule list learning problem that converges to a strict rule list through temperature annealing. NYRULES learns both the discretizations of individual features, and how to compile these features into conjunctive rules without any pre-processing or restrictions. We also learn the rule-order differentiably by introducing a priority score that determines the ordering. NYRULES is able to learn rules of any complexity using specifically optimized predicates and order them in a way that maximizes the predictive performance of the model. As a result, we obtain both highly interpretable, but also accurate rule lists that can assist decision making in a wide range of applications. We demonstrated the effectiveness of NYRULES in extensive real-world and synthetic experiments. We show that NYRULES consistently outperforms both combinatorial and neuro-symbolic methods on a variety of datasets.

**Limitations.** Whilst NYRULES is a powerful tool for interpretable rule learning, it is not without limitations. First and foremost, the rules that NYRULES learns do not allow to draw any causal conclusions about the data generating process without any further assumptions. Thus, they should only be used to assist in decision making and not as a substitute for domain knowledge. Compared to CORELS, we can not give any optimality guarantees on the learned rule list within the search space, but explore a much larger search space that leads to empirically better results. The number of rules in a NYRULES rule list is fixed and must be set beforehand. In our evaluation, we use rule length as a quantitative measure to compare the complexity of rule lists. This choice is supported by early explainability research, which indicates that humans find shorter rules easier to comprehend than longer ones (Huysmans et al., 2011). Nonetheless this proxy is not perfect and does not capture all aspects of interpretability. Hence to undoubtedly determine, which methods is most explainable, a user study would be necessary, which is left for future work. In addition, there are hyperparameters and temperature schedules that need to be set. Whilst we have observed a degree of robustness to these hyperparameters, they require tuning for optimal performance. Lastly, the current rule language is limited to logical conjunctions of thresholded features. Adding disjunctions and more complex logical predicates would be a natural extension of the current work and is something we plan to explore in the future.

**Future Work.** In addition to the expansion of the rule language, there are several other directions in which NYRULES could be extended. For example, the current rule list model is only designed for binary classification tasks, hence to extend it to multi-class classification, is crucial for a wider range of applications. In that context, we also plan to derive a non-conformity score from the rule list model for conformal prediction. Extending NYRULES to regression tasks opens up a wide range of new applications to benefit from interpretable rule lists. Another exciting direction of future work is the adaption of NYRULES to structured data, such as images or graphs. With appropriate predicate functions that extract meaningful concepts in those domains, rule list models could be used as more interpretable and accountable deep learning models.

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

# A  CONVERGENCE OF CONTINUOUS RELAXATIONS

We show that our continuous relaxations for predicate, logical conjunction and rule list converge to their discrete counterparts.

## A.1  PREDICATE.

The soft predicate for a single feature $x_i$ is defined as

$$\hat{\pi}(x_i; \alpha, \beta, t_\pi) = \frac{e^{\frac{1}{t_\pi}(2x_i - \alpha_i)}}{e^{\frac{1}{t_\pi}x_i} + e^{\frac{1}{t_\pi}(2x_i - \alpha_i)} + e^{\frac{1}{t_\pi}(3x_i - \alpha_i - \beta_i)}} .$$

We now show that the soft predicate converges to the hard predicate as $t_\pi \to 0$, which is defined as

$$\pi(x_i; \alpha, \beta) = \begin{cases} 1 & \text{if } x_i \in [\alpha_i, \beta_i] \\ 0 & \text{otherwise} \end{cases} .$$

**Proof:**  *Let us denote as the first logit $a = x_i$, the second logit $b = 2x_i - \alpha_i$ and the third logit $c = 3x_i - \alpha_i - \beta_i$.*

$$\frac{e^{\frac{1}{t_\pi}b}}{e^{\frac{1}{t_\pi}a} + e^{\frac{1}{t_\pi}b} + e^{\frac{1}{t_\pi}c}}$$

$$= \frac{1}{(e^{\frac{1}{t_\pi}a} + e^{\frac{1}{t_\pi}b} + e^{\frac{1}{t_\pi}c}) \cdot e^{-\frac{1}{t_\pi}b}}$$

$$= \frac{1}{e^{\frac{1}{t_\pi}(a-b)} + e^{\frac{1}{t_\pi}(b-b)} + e^{\frac{1}{t_\pi}(c-b)}}$$

$$= \frac{1}{e^{\frac{1}{t_\pi}(a-b)} + 1 + e^{\frac{1}{t_\pi}(c-b)}} .$$

*Consider the following four cases:*

  *1. $x_i < \alpha_i$: Then*

$$b = 2x_i - \alpha_i > 2x_i - x_i = x_i = a ,$$

  *and as then $x_i < \beta_i$, i.e. it is less than the upper bound,*

$$b = 2x_i - \alpha_i > 3x_i - \alpha_i - \beta_i = c .$$

  *Thus $a - b > 0$ and $c - b < 0$, so that in the denominator it holds that in the limit*

$$\lim t_\pi \to 0 \frac{1}{e^{\frac{1}{t_\pi}(a-b)} + 1 + e^{\frac{1}{t_\pi}(c-b)}} = \frac{1}{e^\infty + 1 + e^{-\infty}} = 0 .$$

  *2. $\alpha_i < x_i < \beta_i$: Then*

$$b = 2x_i - \alpha_i > 2x_i - x_i > x_i > a ,$$

  *and as then $x_i \geq \alpha_i$, i.e. it is greater than the lower bound,*

$$c = 3x_i - \alpha_i - \beta_i < 3x_i - \alpha_i - x_i < 2x_i - \alpha_i = b .$$

  *Thus $a - b \leq 0$ and $c - b \leq 0$, so that in the denominator it holds that in the limit*

$$\lim t_\pi \to 0 \frac{1}{e^{\frac{1}{t_\pi}(a-b)} + 1 + e^{\frac{1}{t_\pi}(c-b)}} = \frac{1}{e^{-\infty} + 1 + e^{-\infty}} = 1 .$$

  *3. $x_i = \alpha$ or $x_i = \beta$: Then either $a - b = 0$ and $c - b < 0$, or $a - b > 0$ and $c - b = 0$, so that in the limit*

$$\lim t_\pi \to 0 \frac{1}{e^{\frac{1}{t_\pi}(a-b)} + 1 + e^{\frac{1}{t_\pi}(c-b)}} = \frac{1}{1 + 1 + e^{-\infty}} = 1/2 .$$

  *To obtain the desired behavior at the boundaries, i.e. $\hat{\pi}(x_i) = 1$ or $\hat{\pi}(x_i) = 0$, the output must thus be either ceiled or floored.*

$\square$

## A.2 Logical Conjunction.

The soft logical conjunction for a set of predicates $\hat{\pi}(x_i)$ is defined as

$$\hat{a}(\mathbf{x}) = \frac{\sum_{i=1}^{d} w_i}{\sum_{i=1}^{d} w_i \hat{\pi}(x_i)^{-1}} \ .$$

Given a set of non-negative weights $\mathbf{w} \in [0, \infty)^d$, with at least one weight being positive, the soft logical conjunction takes values in $[0, 1]$ given $d$ predicates $\hat{\pi}(x_i) \in [0, 1]$.

**Proof:** *The domain of the reciprocal is $\hat{\pi}(x_i)^{-1} \in [1, \infty)$. Hence, it holds that all $\forall i \in [d]$ : $w_i \hat{\pi}(x_i)^{-1} \geq w_i > 0$ and thus for their sum $\sum_{i=1}^{d} w_i \hat{\pi}(x_i)^{-1} \geq \sum_{i=1}^{d} w_i > 0$. Then the soft logical conjunction is bounded by*

$$0 \leq \sum_{i=1} \frac{1}{w_i \hat{\pi}(x_i)^{-1}} \leq \frac{\sum_{i=1}^{d} w_i}{\sum_{i=1}^{d} w_i \hat{\pi}(x_i)^{-1}} = \hat{a}(\boldsymbol{x}) \leq 1 \ .$$

*In particular, $\hat{a}(\boldsymbol{x}) = 1$ if $\forall i, w_i > 0 : \hat{\pi}(x_i) = 1$, as then it holds that $\hat{\pi}(x_i)^{-1} = 1$ and $\sum_{i=1}^{d} w_i \hat{\pi}(x_i)^{-1} = \sum_{i=1}^{d} w_i$. On the other hand, $\hat{a}(\boldsymbol{x}) = 0$ if there exists an index $i$ where $w_i > 0$ and $\hat{\pi}(x_i)^{-1} = \infty \leftrightarrow \hat{\pi}(x_i) = 0$.* □

Let us now consider the limit $t_\pi \to 0$. Then, it holds that $\hat{\pi}(x_i) \in \{0, 1\}$ for all $i$. In that case $\hat{a}(\mathbf{x}) \in \{0, 1\}$, as either all $\forall i : \hat{\pi}(x_i) = 1 \lor w_i = 0 \implies \hat{a}(\mathbf{x}) = 1$, or $\exists i : w > 0 \land \hat{\pi}(x_i) = 0 \implies \hat{a}(\mathbf{x}) = 1$, i.e. it corresponds to the logical conjunction of the predicates.

## A.3 Relaxed Conjunction

The relaxed conjunction $\hat{a}(\mathbf{x})$ is defined as

$$\eta = \frac{\epsilon}{\sum_{i=1}^{d} w_i} \ , \quad \hat{a}(\mathbf{x}) = \frac{\sum_{i=1}^{d} w_i}{\sum_{i=1}^{d} w_i \frac{1+\eta}{\hat{\pi}(x_i)+\eta}} \ .$$

We first show that for $\hat{\pi}(x_j) = 0$ the resulting soft conjunction is upper bounded by $\epsilon$.

**Proof:** *Let $\hat{\pi}(x_j) = 0$ and $w_j \geq 1$. Then the relaxed conjunction is*

$$\hat{a}(\boldsymbol{x}) = \frac{\sum_{i=1}^{d} w_i}{\sum_{i \neq j} w_i \frac{1+\eta}{\hat{\pi}(x_i)+\eta} + w_j \frac{1+\eta}{\hat{\pi}(x_i)+\eta}}$$

$$\hat{a}(\boldsymbol{x}) = \frac{\sum_{i=1}^{d} w_i}{\sum_{i \neq j} w_i \frac{1+\eta}{\hat{\pi}(x_i)+\eta} + w_j \frac{1+\eta}{\eta}}$$

*Consider the maximum value of the denominator, i.e. $\hat{\pi}(x_i) = 1$ for all $i \neq j$. Then the denominator is lower bounded by*

$$\sum_{i \neq j} w_i \frac{1+\eta}{\hat{\pi}(x_i)+\eta} + w_j \frac{1+\eta}{\eta} \geq \sum_{i \neq j} w_i \frac{1+\eta}{1+\eta} + w_j \frac{1+\eta}{\eta} = \sum_{i \neq j} w_i + w_j \frac{1+\eta}{\eta} \ .$$

*Thus we have*

$$\hat{a}(\boldsymbol{x}) \leq \frac{\sum_{i=1}^d w_i}{\sum_{i \neq j} w_i + w_j \frac{1+\eta}{\eta}}$$

$$= \frac{\eta \sum_{i=1}^d w_i}{\eta \sum_{i \neq j} w_i + w_j(1 + \eta)}$$

$$= \frac{\eta \sum_{i=1}^d w_i}{\eta \sum_{i=1}^d w_i + w_j}$$

$$= \frac{\frac{\epsilon}{\sum_{i=1}^d w_i} \sum_{i=1}^d w_i}{\frac{\epsilon}{\sum_{i=1}^d w_i} \sum_{i=1}^d w_i + w_j}$$

$$= \frac{\epsilon}{\epsilon + w_j} \leq \epsilon \ .$$

$$\square$$

**Derivatives.** To compute its derivatives, we will use the quotient rule for differentiation, i.e. $\frac{d}{dx} \frac{f(x)}{g(x)} = \frac{f'(x)g(x) - f(x)g'(x)}{g(x)^2}$, where

$$f(\mathbf{x}, \mathbf{w}) = \sum_{i=1}^d w_i \ , \quad \frac{\partial f}{\partial w_j} = 1 \ , \quad \frac{\partial f}{\partial \hat{\pi}(x_j)} = 0$$

$$g(\mathbf{x}, \mathbf{w}) = \sum_{i=1}^d w_i \frac{1+\eta}{\hat{\pi}(x_i) + \eta} \ , \quad \frac{\partial g}{\partial w_j} = \frac{1+\eta}{\hat{\pi}(x_j) + \eta} \ , \quad \frac{\partial g}{\partial \hat{\pi}(x_j)} = -\frac{w_j(1+\eta)}{(\hat{\pi}(x_j) + \eta)^2}$$

Then, the partial derivative of the relaxed conjunction with respect to the predicate $\hat{\pi}(x_j)$ is

$$\frac{\partial \hat{a}(\mathbf{x})}{\partial \hat{\pi}(x_j)} = \frac{0(\sum_{i=1}^d w_i \frac{1+\eta}{\hat{\pi}(x_i)+\eta}) + (\sum_{i=1}^d w_i) \frac{w_j(1+\eta)}{(\hat{\pi}(x_j)+\eta)^2}}{\left(\sum_{i=1}^d w_i \frac{1+\eta}{\hat{\pi}(x_i)+\eta}\right)^2}$$

$$= \frac{(\sum_{i=1}^d w_i) w_j (1+\eta)}{(\hat{\pi}(x_j) + \eta)^2 \left(\sum_{i=1}^d w_i \frac{1+\eta}{\hat{\pi}(x_i)+\eta}\right)^2} \ .$$

Let the predicate $x_l$ be off, i.e. $\hat{\pi}(x_l) = 0$ and $w_l > 0$. Then the derivative for $\hat{\pi}(x_j)$ is

$$\lim_{\hat{\pi}(x_l) \to 0} \frac{\partial \hat{a}(\mathbf{x})}{\partial \hat{\pi}(x_j)} = \frac{(\sum_{i=1}^d w_i) w_j (1+\eta)}{(\hat{\pi}(x_j) + \eta)^2 \left(w_l \frac{1+\eta}{\eta} + \sum_{i \neq j} w_i \frac{1+\eta}{\hat{\pi}(x_i)+\eta}\right)^2} \ .$$

The term $w_l \frac{1+\eta}{\eta}$ is finite and positive, so that the gradient does not vanish if a predicate is off. In the worst case that all predicates are off, i.e. $\hat{\pi}(x_i) = 0$ for all $i$, the derivative is

$$\frac{(\sum_{i=1}^d w_i) w_j (1+\eta)}{\eta^2 \left(\sum_{i=1}^d w_i \frac{1+\eta}{\eta}\right)^2}$$

We note that $1 + \eta \approx 1$ for small $\eta$, so that approximately the gradient is

$$\frac{(\sum_{i=1}^d w_i) w_j}{\eta^2 \left(\sum_{i=1}^d w_i \frac{1}{\eta}\right)^2} = \frac{w_j}{\sum_{i=1}^d w_i} \ .$$

Next, we consider the derivative with respect to the weight $w_j$.

$$\frac{\partial \hat{a}(\mathbf{x})}{\partial w_j} = \frac{1(\sum_{i=1}^{d} w_i \frac{1+\eta}{\hat{\pi}(x_i)+\eta}) - \frac{1+\eta}{\hat{\pi}(x_j)+\eta}(\sum_{i=1}^{d} w_i)}{\left(\sum_{i=1}^{d} w_i \frac{1+\eta}{\hat{\pi}(x_i)+\eta}\right)^2}$$

$$= \frac{\sum_{i=1}^{d} w_i \left(\frac{1+\eta}{\hat{\pi}(x_i)+\eta} - \frac{1+\eta}{\hat{\pi}(x_j)+\eta}\right)}{\left(\sum_{i=1}^{d} w_i \frac{1+\eta}{\hat{\pi}(x_i)+\eta}\right)^2}.$$

Let all predicates be off, i.e. $\hat{\pi}(x_i) = 0$ for all $i$ except for one predicate $\hat{\pi}(x_l) > 0$ with $w_l > 0$. Then the derivative for $w_j$ in the case of $l \neq j$ is

$$\frac{\partial \hat{a}(\mathbf{x})}{\partial w_j} = \frac{w_l\left(\frac{1+\eta}{\hat{\pi}(x_l)+\eta} - \frac{1+\eta}{\eta}\right)}{w_l \frac{1+\eta}{\hat{\pi}(x_l)+\eta} + \sum_{i \neq l}^{d} w_i \frac{1+\eta}{\eta}} > 0,$$

as the numerator the denominator are positive. In the case of $l = j$, the derivative is

$$\frac{\partial \hat{a}(\mathbf{x})}{\partial w_j} = \frac{\sum_{i \neq j} w_i\left(\frac{1+\eta}{\eta} - \frac{1+\eta}{\hat{\pi}(x_j)+\eta}\right)}{w_j \frac{1+\eta}{\hat{\pi}(x_j)+\eta} + \sum_{i \neq j}^{d} w_i \frac{1+\eta}{\eta}}.$$

Again, the numerator is positive as $\frac{1+\eta}{\eta} > \frac{1+\eta}{\hat{\pi}(x_j)+\eta}$ since $\hat{\pi}(x_j) > 0$ whilst the denominator stays the same as in the case of $l \neq j$. Overall, this show that as long as there exists a $\hat{\pi}(x_l) > 0$ with $w_l > 0$, the gradient with respect to the weight $w_j$ is positive.

## A.4 RULE LIST.

The hard rule list $rl(\mathbf{x})$ uses the rule active rule $a_j(\mathbf{x}) = 1$ with the highest priority $p_j$ to determine the output, i.e.

$$rl(\mathbf{x}; \theta, \mathbf{p}) = c_j$$
$$\text{s.t. } a_j(\mathbf{x}; \theta_j) = 1 \ \wedge \ \forall i \neq j : a_i(\mathbf{x}) = 0 \vee p_j > p_i.$$

We defined the soft rule list as

$$\widehat{rl}(\mathbf{x}; \theta, \mathbf{p}) = \sum_{j=1}^{k} c_j \cdot \hat{I}_j(\mathbf{x}),$$

where

$$\hat{I}_j(\mathbf{x}) = e^{\frac{a_j^p(\mathbf{x})+G_j}{t_{rl}}} / \sum_{j=1}^{k} e^{\frac{a_j^p(\mathbf{x})+G_j}{t_{rl}}},$$

where $G_j$ follows a Gumbel distribution, and $a_j^p(\mathbf{x}) = a_j(\mathbf{x}) \cdot p_j$.

Let $t_\pi \to 0$ and $t_{rl} \to 0$, and all priorities be unique, i.e. $\forall j, l : p_j \neq p_k$, and assume that there always is an active rule, i.e. $\forall \mathbf{x} : \exists j : a_j(\mathbf{x}) = 1$, which can be easily achieved by adding an always active rule. Then, the soft rule list converges to the hard rule list.

**Proof:** *Let us consider the limit $t_\pi \to 0$. Then, the soft predicate $\hat{\pi}(x_i; \alpha_i, \beta_i, t_\pi)$ converges to the hard predicate $\pi(x_i; \alpha_i, \beta_i)$ as per A.1 and for all $\forall i : \hat{a}(x_i) \in \{0, 1\}$.*

*We now show that for $t_{rl} \to 0$ the indicator function $\hat{I}_j(\mathbf{x})$ converges to the hard indicator function $I_j(\mathbf{x})$. Given that all priorities are unique, the arg max of $\mathbf{a}(\mathbf{x}) = \mathbf{r}(\mathbf{x}) \cdot \mathbf{p}$ is unique, as either*

$$a_j(\mathbf{x}) = \begin{cases} p_j & \text{if } a_j(\mathbf{x}) = 1 \\ 0 & \text{otherwise} \end{cases}$$

*Since there is at least one active rule with a positive priority $p_j$, the arg max is greater 0 and equal to a $p_j$, which by definition is unique, and which corresponds exactly to the rule for which $a_j(\mathbf{x}) = 1 \wedge \forall i \neq j : a_i(\mathbf{x}) = 0 \vee p_j > p_i$. In the limit of $t_{rl} \to 0$, the Gumbel softmax converges to the arg max function (Jang et al., 2017b), which shows that the soft rule list in the limit is equivalent to the hard rule list.* $\square$

## B  DATASET STATISTICS

We retrieve the datasets from the UCI repository (Dheeru & Efi, 2017), the imodels-data repository (Singh et al., 2021) and the pmlb repository (Romano et al., 2016).

| dataset | #samples | #features | %numerical features | #positive samples | #negative samples | %pos samples |
|---|---|---|---|---|---|---|
| adult | 32561 | 14 | 0.64 | 7841 | 24720 | 0.24 |
| android | 29332 | 86 | 0.00 | 14700 | 14632 | 0.50 |
| breast_cancer | 277 | 17 | 0.06 | 81 | 196 | 0.29 |
| compas_two_year_clean | 6172 | 20 | 0.20 | 2990 | 3182 | 0.48 |
| covid | 1494 | 16 | 0.06 | 809 | 685 | 0.54 |
| credit_card_clean | 30000 | 33 | 0.55 | 6636 | 23364 | 0.22 |
| credit_g | 1000 | 60 | 0.05 | 700 | 300 | 0.70 |
| crx | 690 | 15 | 0.47 | 307 | 383 | 0.44 |
| default | 30000 | 5 | 0.40 | 6636 | 23364 | 0.22 |
| diabetes | 768 | 8 | 1.00 | 268 | 500 | 0.35 |
| eeg_eye_state | 14980 | 14 | 1.00 | 6723 | 8257 | 0.45 |
| electricity | 45312 | 8 | 0.88 | 19237 | 26075 | 0.42 |
| fico | 10459 | 23 | 0.91 | 5459 | 5000 | 0.52 |
| haberman | 306 | 3 | 1.00 | 225 | 81 | 0.74 |
| heart | 270 | 15 | 0.33 | 120 | 150 | 0.44 |
| hepatitis | 155 | 19 | 0.32 | 123 | 32 | 0.79 |
| horse_colic | 368 | 22 | 0.32 | 136 | 232 | 0.37 |
| juvenile_clean | 3640 | 286 | 0.01 | 487 | 3153 | 0.13 |
| madelon | 2600 | 500 | 0.97 | 1300 | 1300 | 0.50 |
| magic | 19020 | 10 | 1.00 | 6688 | 12332 | 0.35 |
| ozone-level | 2534 | 72 | 1.00 | 2374 | 160 | 0.94 |
| pc1 | 1109 | 21 | 1.00 | 77 | 1032 | 0.07 |
| phishing | 11055 | 30 | 0.00 | 6157 | 4898 | 0.56 |
| phoneme | 5404 | 5 | 1.00 | 1586 | 3818 | 0.29 |
| qsar_biodeg | 1055 | 41 | 0.71 | 356 | 699 | 0.34 |
| ring | 7400 | 20 | 1.00 | 3736 | 3664 | 0.50 |
| titanic | 2099 | 8 | 0.38 | 681 | 1418 | 0.32 |
| tokyo1 | 959 | 44 | 0.84 | 613 | 346 | 0.64 |

Table 3: Dataset statistics for the 25 real-world datasets used in our experiments. We say a feature is numerical if it has more than 10 unique values.

## C  HYPERPARAMETERS

For each of the methods we performed a grid search over their hyperparameters and chose the confuirgation that achieved the best performance on the validation datasets: [eeg_eye_state, horse_colic, ozone-level, pc1, breast_cancer] according to the weighted F1 score. For each of the runs we have a time limit of 24 hours, after which the experiments were terminated. The hyperparameters for each of the methods are as follows:

For SBRL, we performed a grid search the following hyperparameters: listlengthprior $\in [2, 3, 4]$; for listwidthprior $\in [1, 2, 3]$; for maxcardinality $\in [1, 2, 3]$ and for minsupport, we fixed the value at 0.05. The number of monte-carlo sampling chains is set to 5 or 10.

For DRNET, we tested the following hyperparameters: lr $\in [0.001, 0.01, 0.1]$; and_lam $\in [0.0001, 0.001, 0.01]$; epochs $\in [1000, 2000, 3000]$; and or_lam $\in [0.0001, 0.001, 0.01]$. As GREEDY has only one hyperparameter, we optimized max_depth $\in [3, 5, 7, 10]$ For CLASSY, we tested the following hyperparameters: beam_width $\in [50, 100, 150, 200]$; n_cutpoints $\in [3, 5, 10]$; and max_depth $\in [3, 5, 10]$. For CORELS, we performed a grid search with the following hyperparameters: c $\in [0.005, 0.01, 0.02]$; n_iter $\in [5000, 10000, 15000]$; max_card $\in [2, 3, 4]$; and min_support $\in [0.01, 0.02, 0.05]$. For NYRULES, we performed a grid search with the following hyperparameters: n_epochs $\in [250, 500, 1000]$; min_support $\in [0.1, 0.2]$; max_support $\in [0.8, 0.9]$; lambd $\in [0.5]$; and lr $\in [0.002, 0.025, 0.05]$. For RLNET, we conducted a grid search with the following hyperparameters: lr $\in [0.001, 0.01, 0.1]$; lambda_and $\in [0.0001, 0.001, 0.01]$; n_epochs $\in [1000, 2000, 3000]$; and l2_lambda $\in [0, 0.001, 0.01]$. To optimize XGBOOST, we explored a range of hyperparameters through grid search: xg_learning_rate $\in [0.001, 0.01, 0.1]$; xg_max_depth $\in [3, 5, 7, 10]$; xg_n_estimators $\in [50, 100, 200, 300]$; and xg_reg_lambda $\in [0, 0.001, 0.01]$.

| | NYRULES | RLNET | RRL | DRNET | GREEDY | CLASSY | CORELS | SBRL | RIPPER | XGBOOST |
|---|---|---|---|---|---|---|---|---|---|---|
| Adult | 0.79 ± 0.01 | 0.81 ± 0.0 | 0.77 ± 0.04 | **0.82** ± 0.01 | 0.80 ± 0.0 | **0.82** ± 0.0 | **0.82** ± 0.0 | 0.65 ± 0.02 | **0.82** ± 0.0 | 0.86 ± 0.0 |
| Android Malware | 0.92 ± 0.0 | **0.95** ± 0.01 | 0.92 ± 0.03 | **0.95** ± 0.01 | 0.87 ± 0.0 | 0.94 ± 0.0 | 0.50 ± 0.0 | $nan \pm nan$ | 0.87 ± 0.03 | 0.96 ± 0.0 |
| COMPAS | 0.66 ± 0.0 | 0.66 ± 0.01 | 0.60 ± 0.02 | 0.64 ± 0.01 | 0.66 ± 0.02 | **0.68** ± 0.02 | 0.65 ± 0.01 | 0.48 ± 0.01 | 0.65 ± 0.01 | **0.68** ± 0.01 |
| Covid ICU | 0.63 ± 0.03 | 0.61 ± 0.05 | 0.63 ± 0.03 | 0.49 ± 0.07 | 0.63 ± 0.02 | 0.61 ± 0.04 | 0.63 ± 0.01 | 0.54 ± 0.03 | **0.64** ± 0.02 | **0.64** ± 0.02 |
| Credit Card | **0.82** ± 0.0 | 0.81 ± 0.01 | 0.75 ± 0.07 | 0.80 ± 0.01 | **0.82** ± 0.0 | 0.81 ± 0.01 | **0.82** ± 0.0 | 0.52 ± 0.02 | 0.78 ± 0.01 | **0.82** ± 0.01 |
| German Credit | **0.72** ± 0.03 | **0.72** ± 0.02 | **0.72** ± 0.03 | 0.30 ± 0.02 | **0.72** ± 0.04 | 0.71 ± 0.04 | 0.70 ± 0.01 | 0.70 ± 0.02 | **0.72** ± 0.05 | 0.75 ± 0.02 |
| Credit Screening | **0.86** ± 0.02 | 0.84 ± 0.03 | 0.82 ± 0.03 | 0.50 ± 0.05 | **0.86** ± 0.02 | 0.85 ± 0.02 | 0.74 ± 0.04 | **0.86** ± 0.02 | **0.86** ± 0.02 | 0.85 ± 0.02 |
| Diabetes | 0.73 ± 0.02 | 0.73 ± 0.03 | **0.75** ± 0.05 | 0.45 ± 0.12 | 0.72 ± 0.03 | 0.73 ± 0.04 | 0.73 ± 0.03 | 0.55 ± 0.11 | **0.75** ± 0.06 | 0.74 ± 0.02 |
| Electricity | **0.76** ± 0.0 | 0.71 ± 0.01 | 0.65 ± 0.08 | 0.67 ± 0.01 | **0.76** ± 0.0 | 0.66 ± 0.0 | 0.73 ± 0.01 | 0.53 ± 0.02 | **0.76** ± 0.01 | 0.84 ± 0.0 |
| FICO | **0.70** ± 0.01 | 0.68 ± 0.01 | 0.65 ± 0.02 | 0.64 ± 0.02 | **0.70** ± 0.01 | 0.68 ± 0.02 | 0.65 ± 0.01 | 0.52 ± 0.01 | **0.70** ± 0.01 | 0.72 ± 0.01 |
| Heart Disease | 0.79 ± 0.04 | 0.75 ± 0.01 | 0.72 ± 0.04 | 0.52 ± 0.11 | 0.71 ± 0.04 | 0.78 ± 0.09 | 0.69 ± 0.05 | 0.61 ± 0.15 | **0.80** ± 0.05 | 0.79 ± 0.08 |
| Hepatitis | 0.79 ± 0.06 | 0.79 ± 0.06 | 0.79 ± 0.07 | 0.24 ± 0.05 | 0.80 ± 0.05 | 0.78 ± 0.03 | **0.83** ± 0.04 | 0.79 ± 0.04 | 0.78 ± 0.07 | **0.83** ± 0.06 |
| Juvenile | 0.88 ± 0.02 | 0.89 ± 0.01 | 0.88 ± 0.01 | **0.90** ± 0.01 | 0.86 ± 0.01 | 0.89 ± 0.01 | 0.87 ± 0.01 | $nan \pm nan$ | 0.13 ± 0.01 | **0.90** ± 0.01 |
| Magic | **0.80** ± 0.01 | 0.79 ± 0.01 | 0.74 ± 0.02 | 0.79 ± 0.02 | 0.74 ± 0.01 | 0.77 ± 0.01 | 0.74 ± 0.0 | 0.57 ± 0.05 | 0.78 ± 0.01 | 0.87 ± 0.0 |
| Phishing | 0.91 ± 0.01 | 0.93 ± 0.01 | 0.83 ± 0.06 | **0.94** ± 0.0 | 0.89 ± 0.0 | 0.92 ± 0.01 | 0.44 ± 0.01 | 0.87 ± 0.02 | 0.89 ± 0.0 | 0.95 ± 0.0 |
| Phoneme | 0.78 ± 0.02 | 0.74 ± 0.01 | 0.74 ± 0.02 | 0.74 ± 0.01 | 0.76 ± 0.01 | **0.81** ± 0.01 | 0.75 ± 0.01 | 0.70 ± 0.04 | 0.77 ± 0.02 | 0.88 ± 0.01 |
| QSAR | 0.81 ± 0.03 | **0.84** ± 0.01 | 0.80 ± 0.02 | 0.64 ± 0.02 | 0.74 ± 0.03 | 0.82 ± 0.03 | 0.74 ± 0.01 | 0.71 ± 0.02 | 0.79 ± 0.03 | 0.86 ± 0.02 |
| Ring | **0.92** ± 0.02 | 0.81 ± 0.01 | 0.83 ± 0.04 | 0.50 ± 0.02 | 0.61 ± 0.02 | 0.68 ± 0.02 | 0.66 ± 0.02 | 0.70 ± 0.02 | 0.75 ± 0.04 | 0.94 ± 0.0 |
| Titanic | **0.79** ± 0.02 | 0.77 ± 0.02 | 0.72 ± 0.05 | 0.45 ± 0.08 | **0.79** ± 0.02 | **0.79** ± 0.02 | 0.71 ± 0.03 | 0.32 ± 0.02 | 0.78 ± 0.02 | 0.81 ± 0.03 |
| Tokyo | 0.91 ± 0.03 | 0.91 ± 0.02 | 0.91 ± 0.01 | 0.37 ± 0.05 | 0.88 ± 0.01 | **0.92** ± 0.02 | 0.87 ± 0.03 | **0.92** ± 0.01 | **0.92** ± 0.03 | 0.93 ± 0.02 |
| Rank | **3.25** | 3.98 | 5.78 | 6.55 | 4.50 | 3.70 | 5.83 | 7.31 | 3.95 | $n/a$ |

Table 4: We report the results comparison on 20 real world datasets stemming from domains such as medicine, finance, and criminal justice. We compare NYRULES against CORELS, SBRL, CLASSY, GREEDY, RLNET, RRL, DRNET, and XGBOOST. We report the accuracy averaged over 5-fold cross validation. The experiments were terminated after 24 hours, indicated by $n/a$. NYRULES performs the best with respect to the $Acc$ score, indicated by the lowest rank.

## C.1 TEMPERATURE SCHEDULES

Temperature schedules are crucial in optimization problems involving soft approximations of discrete functions. They help in gradually transitioning from a soft to a hard decision boundary, which can improve convergence and performance. By adjusting the temperature parameter, we control the smoothness of the approximations, allowing the model to explore the solution space more effectively during the initial stages of training and then refine the solutions as training progresses.

We use a linear temperature decay during the second half of training for both temperatures. The temperature starts at $1.0$ and linearly decreases to $0.1$ for the rule priority temperature $t_{rl}$ and ranges from $0.2$ to $0.05$ for the predicate temperature $t_{\pi}$. These values were determined through hyperparameter optimization and are unchanged across all experiments. The temperature is updated at each epoch as follows:

```
temp_start = 1.0
temp_end = 0.1
temp = temp_start
step_size = (temp_start - temp_end)/(total_epochs*2)
If epoch >= total_epochs/2:
    temp = temp - step_size
```

This schedule allows the model to maintain a high level of flexibility during the first half of training with multiple active rules and gradually focus on only a single rule per sample in the latter half.

## D REAL WORLD: ACCURACY

We report the accuracy of the methods on the real world datasets in Table 4. The conclusions about the performance of the methods are consistent with the results obtained using the weighted F1 score. Although, the improvements of NYRULES over the baselines is more pronounced in terms of accuracy. Nonetheless, we report the weighted F1 score as the main evaluation metric, as it is more informative about the performance of the methods in the presence of class imbalance.

## E SYNTHETIC DATA

Lastly, we compare the best performing rule list models under different challenging settings. We sample $d$ independent, uniform feature variables $X_i$ $n$ times $\{\mathbf{x}_i \mid \mathbf{x}_i \sim \mathcal{U}(0,1)^d\}$. We sample $m$ indices $Ind = \{ind_1, \ldots, ind_m\}$ out of $\{1, \ldots, d\}$ to be included in the rule. The range of each

| Dataset | F1 | Ablation | Diff |
|---|---|---|---|
| adult | 0.80 | 0.66 | 0.14 |
| android | 0.92 | 0.33 | 0.59 |
| breast-cancer | 0.69 | 0.62 | 0.07 |
| compas-two-year-clean | 0.66 | 0.35 | 0.31 |
| covid | 0.62 | 0.29 | 0.34 |
| credit-card-clean | 0.79 | 0.68 | 0.11 |
| credit-g | 0.72 | 0.14 | 0.58 |
| crx | 0.86 | 0.58 | 0.28 |
| diabetes | 0.73 | 0.51 | 0.22 |
| electricity | 0.75 | 0.42 | 0.33 |
| fico | 0.70 | 0.31 | 0.39 |
| haberman | 0.69 | 0.70 | -0.01 |
| heart | 0.78 | 0.40 | 0.39 |
| hepatitis | 0.79 | 0.07 | 0.71 |
| juvenile-clean | 0.88 | 0.80 | 0.07 |
| magic | 0.79 | 0.51 | 0.28 |
| phishing | 0.91 | 0.78 | 0.12 |
| phoneme | 0.79 | 0.59 | 0.20 |
| qsar-biodeg | 0.81 | 0.53 | 0.28 |
| ring | 0.92 | 0.33 | 0.59 |
| titanic | 0.77 | 0.54 | 0.23 |
| tokyo1 | 0.91 | 0.19 | 0.72 |

interval $\beta - \alpha$ is determined by the target rule fraction $s$ and the number of predicates as

$$r = \beta - \alpha = s^{\frac{1}{m}}\ .$$

We thus create randomly uniform intervals by sampling for each feature $i \in Ind$ a lower bound $\alpha_i \sim \mathcal{U}(0, 1 - r)$ and corresponding upper bound $\beta_i = \alpha_i + r$. These intervals are then combined into a rule $a_j(\mathbf{x}) = \bigwedge_{i \in Ind} \alpha_i < x_i < \beta_i$ that covers on expectation $s$ of the total datapoints. We repeat this process to generate $k$ rules. Each rule is assigned a random priority $p_j \sim [1, \ldots, k]$. We set $c_j = 1$ or $c_j = 0$ uniformly at random to determine each rules class label. Thus, abiding by the corresponding rule list $rl(\mathbf{x})$, we assign the class label $y_i = c_j$ for the rule $j$ where $a_j(\mathbf{x}) = 1$ and $p_j$ is maximal. Finally, for all samples $\mathbf{x}_i$ where $\forall j : a_j(\mathbf{x}) = 0$, we assign the class label $y_i = 0$ with probability $0.5$ and $y_i = 1$ otherwise.

We use the following parameters to generate the datasets reported in the experiments:

1. Rule complexity: $d = 20, n = 5000, s = 0.1, k = 2, m \in \{2, 4, 6, 8\}$.
2. Number of rules: $d = 20, n = 5000, s = 0.1, k = \{2, 4, 6, 8, 12\}, m = 2$.
3. Sample complexity: $d = 20, n \in \{100, 500, 1000, 5000, 10000\}, s = 0.1, k = 2, m = 2$.

**Rule Complexity.** We begin by varying rule complexity through increasing the number of predicates $\pi_i$ per rule. We report the $F_1$ score in Figure 7a. Overall, NYRULES is consistently the best performing method. In particular, NYRULES is the only method that does not require prediscretization of the features and can learn the thresholds of the predicates exactly. Especially for

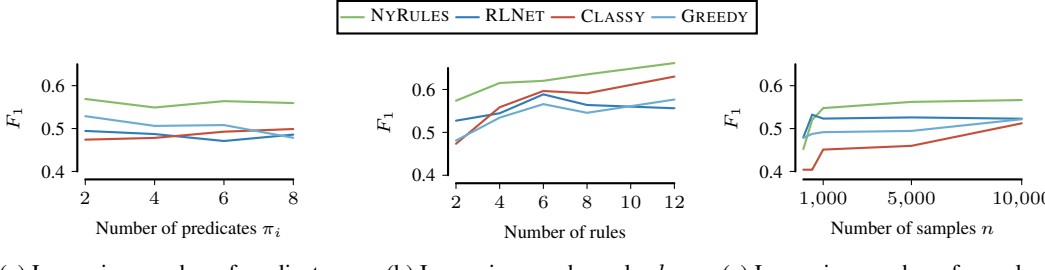

(a) Increasing number of predicates.  (b) Increasing number rules $k$.  (c) Increasing number of samples.

Figure 7: $F_1$ score of a synthetic rule list. NYRULES is performant for complex rules **(a)**, rule lists with few and many rules **(b)** and scales well with a large number of samples **(c)**.

| | Uniform | Diff | kMeans | Diff | Fixed **a** | Diff | Hard Conj. | Diff |
|---|---|---|---|---|---|---|---|---|
| heart | 0.78 | -0.00 | 0.79 | 0.01 | 0.79 | 0.01 | 0.40 | -0.39 |
| credit-g | 0.66 | -0.06 | 0.71 | -0.01 | 0.64 | -0.09 | 0.14 | -0.58 |
| juvenile-clean | 0.89 | 0.01 | 0.86 | -0.02 | 0.80 | -0.07 | 0.80 | -0.07 |
| compas-two-year-clean | 0.59 | -0.06 | 0.65 | -0.00 | 0.66 | 0.00 | 0.35 | -0.31 |
| fico | 0.65 | -0.05 | 0.70 | -0.00 | 0.68 | -0.02 | 0.31 | -0.39 |
| credit-card-clean | 0.79 | -0.00 | 0.72 | -0.07 | 0.68 | -0.11 | 0.68 | -0.11 |
| android | 0.92 | -0.00 | 0.92 | 0.00 | 0.92 | -0.00 | 0.33 | -0.59 |
| phishing | 0.91 | 0.00 | 0.91 | 0.00 | 0.91 | 0.00 | 0.78 | -0.12 |
| electricity | 0.63 | -0.12 | 0.74 | -0.00 | 0.74 | -0.01 | 0.42 | -0.33 |
| qsar-biodeg | 0.79 | -0.02 | 0.79 | -0.02 | 0.79 | -0.02 | 0.53 | 0.28 |
| phoneme | 0.66 | -0.13 | 0.77 | -0.02 | 0.59 | -0.20 | 0.59 | -0.20 |
| adult | 0.75 | -0.04 | 0.66 | -0.14 | 0.66 | -0.14 | 0.66 | -0.14 |
| covid | 0.61 | -0.02 | 0.64 | 0.02 | 0.64 | 0.02 | 0.29 | -0.34 |
| diabetes | 0.73 | 0.00 | 0.74 | 0.00 | 0.66 | -0.07 | 0.51 | -0.22 |
| hepatitis | 0.79 | -0.00 | 0.76 | -0.02 | 0.76 | -0.03 | 0.07 | -0.71 |
| magic | 0.75 | -0.04 | 0.75 | -0.04 | 0.77 | -0.02 | 0.51 | -0.28 |
| titanic | 0.77 | -0.01 | 0.77 | -0.01 | 0.77 | -0.01 | 0.54 | -0.23 |
| tokyo1 | 0.91 | 0.00 | 0.91 | 0.00 | 0.91 | 0.00 | 0.19 | -0.72 |
| crx | 0.85 | -0.00 | 0.86 | 0.00 | 0.86 | -0.00 | 0.58 | -0.28 |
| ring | 0.73 | -0.19 | 0.72 | -0.20 | 0.80 | -0.12 | 0.33 | -0.59 |
| Average | 0.76 | -0.04 | 0.77 | -0.03 | 0.75 | -0.04 | 0.43 | -0.3 |

Table 5: Ablation study comparing the obtained $F_1$ scores using uniform and kmeans based binning, as well as a fixed rule priority. NYRULES accuracy is negatively impacted by each's components removal.

the most complex rules with 8 predicates, NYRULES holds the biggest advantage over the other methods. Here, exact thresholding is crucial, as inaccuracies over multiple dimensions compound and limit the obtainable $F_1$ score with pre-discretized features.

**Number of Rules.** Next, we increase the number of rules $k$ sampled in a rule list, whilst keeping each rule fixed at 2 predicates. We report the $F_1$ score in Figure 7b. NYRULES maintains a consistent advantage over it competition in both the low and high rule list regime.

**Sample Complexity.** Lastly we examine the sample complexity of each method, reported in Figure 7c. For 100 sampled datapoints, all methods are closely matched. NYRULES gradient based optimization continuously improves as the number of samples increases, whilst RLNET and GREEDY improvement caps out at 1000 samples. In the high sample regime of 10.000 samples, NYRULES holds the biggest advantage over the other methods.

The synthetic experiments further highlight the advantage of fully flexible thresholding, especially for complex rules, and show that NYRULES scales well with the number of samples and rules. Therefore, in real world applications where the majority of features are continuous and with many samples, NYRULES can offer a competitive edge over existing alternatives.

# F ABLATION STUDIES

We perform an ablation study to investigate the impact of the different components of our method. We provide the $F_1$ score when using *uniform*, *kmeans* pre-processing of continuous features, a fixed rule priority **p** and with a hard conjunction in Table 5.

