# OpenReview forum: "Neuro-Symbolic Rule Lists"
_ICLR.cc/2025/Conference — Submitted to ICLR 2025_

### Official Review · Reviewer_7NXf · 2024-10-30

**Soundness:** 3
**Presentation:** 3
**Contribution:** 3
**Rating:** 6
**Confidence:** 4

**Summary:**

This paper proposes a new framework for learning rule lists by gradient descent, named NyRules. First, the authors introduce a differentiable representation of a rule list using some existing techniques, such as the Gumbel-Softmax. Then, they formulate the task of learning a rule list that minimizes the empirical risk and the constraints on the support as a continuous optimization problem. One of the advantages of the proposed method is that it does not require pre-processing steps such as discretization and rule mining, which are required by most existing methods and often incur additional computational costs. Experimental results demonstrated that the proposed method achieved good accuracy compared to the baselines across all the datasets.

**Strengths:**

Overall, I like this paper, and I think the authors make solid contributions to the community on interpretable machine learning.
- S1. This paper is well-written, well-organized, and easy to follow.
- S2. The authors introduce a differentiable representation of rule lists, which is a novel approach to the best of my knowledge. While the proposed representation mainly consists of existing methods, including [Yang et al. 2018], [Xu et al. 2024], and [Jang et al. 2017], it also includes some unique techniques, such as relaxed formulation for alleviating the vanishing gradients issues. In addition, I think the illustrative demonstration of each continuous relaxation (Figures 3 and 4) helps readers understand the effectiveness of the proposed approach well.
- S3. The experimental results demonstrated that the proposed method often achieved better accuracy than the existing methods. The proposed method also attained comparable accuracy to XGBoost, which was surprising to me.

**Weaknesses:**

While I think this is a good paper, I believe the following concerns should be addressed to improve the quality of this paper:
- W1. One of my major concerns is the computational cost of the proposed method. I could not find the experimental results on the running times of the proposed method and baselines, even in the appendix. To claim that the proposed method overcomes the scalability issue of the existing methods, I think the running time of the proposed method should be compared with the existing methods. Although I agree that it is a desirable advantage that the proposed method does not require discretization and rule mining as pre-processing, I am worried about the scalability of the proposed method because its average running time was not reported in the paper.
- W2. This paper looks to be missing descriptions of some important information. For example, this paper seems to assume a binary classification task implicitly. I think it should be explicitly mentioned at, for example, the beginning of Section 2. In addition, as mentioned above, this paper seems to lack the experimental results on the computational times of the proposed method and baselines. I think such information should be reported because scalability is one of the important factors for practitioners when deciding which algorithm they should use for their task.

**Questions:**

- Q1. Could you show the experimental results on the average running times of the proposed method to the baselines?
- Q2 (optional). From the objective function in Section 3.4, the proposed method does not explicitly impose constraints on the sparsity of predicates in each rule. However, from Figure 5(b), the experimental results suggest that the proposed method tends to obtain sparse rules, which was surprising to me. Did you carry out any post-processing for it? In addition, I guess that the required length of each rule to maintain sufficient accuracy may vary depending on the dataset. Does the trend in Figure 5(b) differ depending on the dataset?
- Q3 (optional). In the learning problem of the proposed method, the constraints on minimum and maximum supports are relaxed as soft constraints (i.e., regularizer). In the experiments, how often did the learned rule lists violate the constraints on minimum and maximum supports? Also, while the learned rule lists may not violate this constraint if we set $\lambda$ to be a sufficiently large value, how does it affect the accuracy or computational cost of the proposed method?
- Q4 (optional). If I understand correctly, the proposed learning algorithm gradually decreases the annealing temperatures $t_{\pi}$ and $t_{rl}$. Could you show the pseudo-code of a procedure for determining the values of $t_{\pi}$ and $t_{rl}$ in each step during training?

---

> ### Author Response · Authors · 2024-11-18
>
> Dear Reviewer,
> We thank you for your detailed review and would like to address your concerns and suggestions.
> - **Runtime**: We provide the average runtimes of all methods on the real-world datasets below. NyRules is faster than DRS, RLNet, and SBRL slower than the greedy approaches and RRL. In general, NyRules incurs a computational overhead compared to the greedy methods but compensates for it in terms of classification accuracy. Improving the scalability of NyRules for large-scale settings is an exciting direction for future work. We will add the runtimes and discuss them in the updated manuscript.
> | Method     |      Avg. Time(s) |
> |:-----------|--------:|
> | GREEDY    |    0.05 |
> | CORELS    |    1.04 |
> | RRL       |    1.53 |
> | CLASSY     |    6.96 |
> | **NyRules** |   75.42 |
> | DRS     |  144.88 |
> | RLNET     |  222.96 |
> | SBRL      | 1489.23 |
> - **Multi-class classification**: We focus on differentiably learning the rules and their order, and less so on the consequents. As many rule list methods only support binary classification (SBRL, CORELS, DRS), we evaluated exclusively binary classification. We can extend NyRule to multi-class settings by expanding the dimension of the consequent vector to $l$ where $l$ is the number of classes. We provide preliminary results below on sample datasets, where we fit rule lists of length $n_rules = l*5$. Here, NyRules also outperforms the SOTA.
> | Dataset      | NyRules    | RLNet        | CLASSY   | GREEDY   |
> |:-------------|:--------------|:--------------|:---------|:----------|
> | car          | 0.83          | **0.84** | 0.83     | 0.05      |
> | ecoli        | **0.84** | 0.78          | 0.75     | 0.55      |
> | iris         | **0.95** | 0.83          | 0.94     | 0.56      |
> | yeast        | **0.54** | 0.41          | 0.52     | 0.21      |
> | Avg.~Rank    | **1.50** | 2.25          | 2.25     | 4.00      |
> - **Rule sparsity**: We did not carry out post-processing when assessing rule sparsity; we simply counted all predicates with $w_i>0$. The sparsity can be partially attributed to the minimum support regularizer. Rules that are too complex, i.e., impose too many conditions, will likely have very low or even zero coverage. Therefore, this implicitly guides NyRules to learn rules with fewer conditions.
> - **Constraint violation**: During training, we generally observed that the learned rule list adhered to the min-/max-support constraints imposed by the regularizer. However, we cannot provide concrete statistics for this observation as we did not track the magnitude of the regularization term during training.
> - **Temperature schedule**: We use a linear temperature decay during the second half of training for both temperatures. We will include a description of temperature scheduling in the updated Appendix.
> ```
> temp_start = 1.0
> temp_end = 0.1
> temp = temp_start
> step_size = (temp_start - temp_end)/(total_epochs*2)
> If epoch >= total_epochs/2:
> 	temp = temp - step_size
> ```
> We hope this adequately addresses your concerns and questions, and we would be happy to discuss further questions. We will incorporate the proposed changes within a few days into the manuscript and notify you once the updated version is available.

---

> > ### Author Response · Authors · 2024-11-25
> >
> > Dear Reviewer, the updated manuscript is now available on OpenReview where we have incorporated your valuable feedback. We look forward to your response.

---

> > > ### Comment · Reviewer_7NXf · 2024-11-26
> > >
> > > Dear Authors, I would like to thank the authors for their insightful responses and revised manuscript. Since they addressed some of my concerns, I decided to keep my score positive. (*Additional comment:* I guess the average running time of each method varies depending on the size of each dataset. So if the paper is accepted, it may be better to show the running time for each dataset, rather than averaging over all the datasets. )

---

> > > > ### Author Response · Authors · 2024-11-26
> > > >
> > > > Thank you for your feedback and positive assessment of our revisions. We appreciate your suggestion about presenting running times per dataset and will incorporate it in the final version of the paper.
> > > >
> > > > To further improve the manuscript, could you let us know if there’s anything else we can address to strengthen the paper and potentially improve the score?

---

### Official Review · Reviewer_yZdc · 2024-10-30

**Soundness:** 2
**Presentation:** 3
**Contribution:** 2
**Rating:** 3
**Confidence:** 5

**Summary:**

The paper presents a rule learning system for numerical data, that integrates discretization and learning. This is, by itself, not a novel contribution. Classic rule learning algorithms such as Ripper can do that as well. It also optimizes the rule order of the resulting list, which is a somewhat debatable contribution, because rules in a decision list are not independent of the order. This is also a problem for interpretability, BTW. Results are presented in F1-scores, which is somewhat unconventional for work in this area, and makes a comparison harder. I also miss a comparison to the (still) standard benchmark Ripper, in the Java implementation available, e.g., in Weka (this is important, the Python implementation Wittgenstein is considerably worse, and unfortunately many of the published claims of superiority over Ripper are based on that implementation). As a result, I find it hard to assess the value of this work. I do think that the paper (although generally well written) could be much stronger and more credible with respect to the experimental evaluation.

**Strengths:**

- nice formal presentation of the work

**Weaknesses:**

- experimental results are somewhat non-standard, in particular the use of F1. Why do you think F1 is a good measure for evaluating classification problems? It is quite adequate for information retrieval, because there is an asymmetry between the positive and negative class, so that it is o.k. that unretrieved irrelevant documents do not influence the result, but I don't see why ignoring true negatives should be a good idea in any of the problems studied here.

- The results also seem to be flawed at times. For example, results of CORELS at 0.33 or 0.27 when other algorithms are above 0.9 (similar for SBRL) indicate for me that the algorithms have not been adequately used. Possible explanations could be that they did not learn any rules, or learned for the wrong class, or similar. In my view, no algorithm can be as bad as 0.16 for Titanic. I guess you could invert its predictions and would get a better result (this would be clearer if accuracy had been used as an evaluation metric).

- The authors seem to assume that decision lists are just an ordering of rules. Of course, an ordered rule set is a decision list, but the opposite is not true. This affects optimality and interpretability. Consider, for example the decision list
IF A then +
ELSIF B then -
ELSIF C then +
ELSE -
Note that each rule is only valid in context. It is, e.g., not true that all C's are positive, only if they are not B.
If you simply reorder the rules in the decision list (e.g., by swapping the two positive rules) this may drastically change the semantics. This is why classical rule learning algorithms like Ripper always learn a decision list in Sequence (Ripper does optimize it later on).

On the other hand, you could also rewrite the decision list as a rule set as follows:
IF A then +
IF B and NOT A then -
IF C and NOT B and NOT A then +
In that case, you could reorder the rules as you wish. If rules like this are learned, they would be longer (which would be consistent with what the authors observed on their algorithm), but, on the other hand, the ordering is practically irrelevant.

So probably, the authors learn a mixture of both (optimizing both the order and the rule bodies at the same time), but this should be discussed deeper.

- The ordering also has an effect on interpretability, of course. A long decision list cannot be easily interpreted, because the rules cannot be interpreted independently, you always have to interpret them in context of all previous rules. So the claim already posed in the introduction that decision lists are interpretable is debatable, in particular for long lists. It does hold if rules can be interpreted independently, but in that case the order would be irrelevant (see above). The authors should more clearly discuss the trade-off between ordered rules and interpretability.

UPDATE: Based on the authors' response, I have downgraded my evaluation. They provide results for Ripper, but they seem to be flawed or at least not reproducible with the provided information. See my reply for details.

**Questions:**

Is there a reference for GREEDY? What implementations of the algorithms did you use?

---

> ### Author Response · Authors · 2024-11-18
>
> Dear Reviewer,
> We thank you for your detailed review and would like to address your concerns and suggestions.
>
> - **Rule ordering**: Indeed, the order of a set of overlapping rules is very impactful. RIPPER, for example, first learns rules and then their order but does not update the rule bodies. On the other hand, because rule order is very important and influences the resulting classifier, NyRules jointly optimizes rule bodies and order. We study the effect of learning a dynamic rule order instead of a static one in an additional ablation experiment. On average, the performance decreases by 0.04 F1 points. The decrease is correlated with the complexity of the dataset, where more performance is lost on the more complex datasets (Adult, credit card, juvenile).
> | Dataset               |   F1 with Fixed Order |   Difference |
> |:----------------------|----------------------:|-------------:|
> | adult                 |                  0.66 |        -0.14 |
> | android               |                  0.92 |        -0    |
> | compas_two_year_clean |                  0.66 |         0    |
> | covid                 |                  0.64 |         0.02 |
> | credit_card_clean     |                  0.68 |        -0.11 |
> | credit_g              |                  0.64 |        -0.09 |
> | crx                   |                  0.86 |        -0    |
> | diabetes              |                  0.66 |        -0.07 |
> | electricity           |                  0.74 |        -0.01 |
> | fico                  |                  0.68 |        -0.02 |
> | heart                 |                  0.79 |         0.01 |
> | hepatitis             |                  0.76 |        -0.03 |
> | juvenile_clean        |                  0.8  |        -0.07 |
> | magic                 |                  0.77 |        -0.02 |
> | phishing              |                  0.91 |         0    |
> | phoneme               |                  0.59 |        -0.2  |
> | qsar_biodeg           |                  0.79 |        -0.02 |
> | ring                  |                  0.8  |        -0.12 |
> | titanic               |                  0.77 |        -0.01 |
> | tokyo1                |                  0.91 |         0    |
> | Average               |                  0.75 |        -0.04 |
> - **Comparison to Ripper**: We set up a comparison against RIPPER using Weka's default setting. We set the minimum support of each rule to 5% of total samples to limit the theoretical maximum size of the rule list to 20 since there exists no max_rules parameter. Otherwise, RIPPER learns rule lists with up to 100 rules. Both the F1 scores and accuracy are reported below. RIPPER is competitive with NyRules and beats it 25% of the time when measuring F1 and 20% of the time using Accuracy. We will include RIPPER as a competitor in the Experiment section.
> | Dataset               |   RIPPER F1 |   NyRules F1 |   RIPPER Accuracy |   NyRules Acc |
> |:----------------------|------------:|-------------:|------------------:|--------------:|
> | heart                 |        0.74 |         0.78 |              0.75 |          0.79 |
> | credit_g              |        0.7  |         0.72 |              0.72 |          0.72 |
> | compas_two_year_clean |        0.65 |         0.66 |              0.65 |          0.66 |
> | fico                  |        0.7  |         0.7  |              0.7  |          0.7  |
> | credit_card_clean     |        0.76 |         0.79 |              0.78 |          0.82 |
> | android               |        0.87 |         0.92 |              0.87 |          0.92 |
> | phishing              |        0.89 |         0.91 |              0.89 |          0.91 |
> | electricity           |        0.77 |         0.75 |              0.77 |          0.76 |
> | qsar_biodeg           |        0.79 |         0.81 |              0.8  |          0.81 |
> | phoneme               |        0.77 |         0.79 |              0.77 |          0.78 |
> | adult                 |        0.8  |         0.8  |              0.82 |          0.79 |
> | covid                 |        0.64 |         0.62 |              0.64 |          0.63 |
> | diabetes              |        0.74 |         0.73 |              0.74 |          0.73 |
> | hepatitis             |        0.74 |         0.79 |              0.76 |          0.79 |
> | magic                 |        0.77 |         0.79 |              0.78 |          0.8  |
> | titanic               |        0.76 |         0.77 |              0.78 |          0.79 |
> | tokyo1                |        0.91 |         0.91 |              0.91 |          0.91 |
> | crx                   |        0.86 |         0.86 |              0.86 |          0.86 |
> | ring                  |        0.73 |         0.92 |              0.74 |          0.92 |
> | Win percentage           |        25% |         75% |              20%  |          80%  |

---

> ### Author Response · Authors · 2024-11-18
>
> - **$F_1$-score**: Accuracy and F1-score are widely accepted metrics for assessing classification performance. We provide that class-weighted F1 score, which is the F1 for each class weighted by the class frequency, in the main paper and the accuracy in the Appendix.
> - **Performance of baselines**: We optimized the hyperparameters of each method for maximum F1-score on 5 separate holdout datasets. Specific hyperparameter tuning on each dataset may improve CORELS and SBRL performance. However, on Titanic, for example, SBRL achieves an accuracy of 0.31 (see Appendix D), which indicates that the optimization failed severely but not that a good model can be obtained through inversion.
> - **Implementations**: We use the source code provided by the authors of RLNET, RRL, DRNET, and CLASSY. We use the official XGBoost Python package. For CORELS and CLASSY, we use the Python3-compatible implementation provided by the imodels Python package. The GREEDY method used in the paper also comes from the imodels package. It learns a rule list greedily and iteratively. We will add this information to the updated manuscript.
> - **Interpretability**: We agree that each nested rule must be interpreted in the context of the previous rules, which is a feature in rule lists in general. We will update the text to reflect this accurately.
>
> We hope this adequately addresses your concerns and questions, and we would be happy to discuss further questions. We will incorporate the proposed changes within a few days into the manuscript and notify you once the updated version is available.

---

> > ### Author Response · Authors · 2024-11-25
> >
> > Dear Reviewer, the updated manuscript is now available on OpenReview where we have incorporated your valuable feedback. We look forward to your response.

---

> ### Comment · Reviewer_yZdc · 2024-11-27
>
> I am afraid I am not all too happy with your response.
>
> *Ripper and Order*: It is not correct that Ripper learns an order (it does not, it learns rules in order and keeps that order) and it is not correct that it does not update the rule bodies. It also aims at optimizing order and sequence in a post-processing phase, which keeps the order fixed, but tries to re-learn rules in the context of other rules (prior rules and subsequent rules).  In a way, this is the opposite of what you claim that it does (it does not keep the rule bodies fixed and change the order, but it keeps the order fixed and changes the rule bodies in its optimization phase after the initial learning of the rules).
>
> *Evaluation*: The Ripper results did not make it into the paper or the appendix? I can't seem to find it in the uploaded version.
> It is also unclear how you compute the win percentage. In accuracy, Ripper wins 4 out of 19, if I counted correctly, NyRules wins 11 of 19, with 4 ties. So even if you count all of the ties as losses it is 21%.
> On the other hand, in F1, I count only 3 wins for Ripper, 4 ties, and 12 for NyRules. That seems worse than in accuracy? Maybe if you provide more than two digits this would clarify.
>
> Moreover, I have tried hepatitis in Weka with a 5-fold cross-validation as you specified in the paper, and I get for the default parameters with three different seeds (1, 2, and 3): 82.58%, 80.00%, 82.56%. These are all considerably higher than the 0.76 accuracy that you report. Note that the results for a 10-fold x-val in Weka are lower. But this is not what you are doing according to to the paper. Also for adult (where ripper is already better), I get 82.9% (for seed 1, tried only one here as it is big), and for credit_g I get 69%, 72.1% and 71%. Two of the last are worse than what you report but that also shows how brittle the results of a 5-fold X-val are on such small datasets. I did not try any other than these three datasets.
>
> My results are without parameter optimization (I understand from your reply that you did a parameter optimization on 5 separate hold-out sets, so if you let me know the parameter you selected for Ripper, I can double-check that).
>
> *F1*: I don't agree that F1 is a good choice for classification performance, and I also don't think it is widely used for such tasks. For example, one of the largest studies in that area (https://jmlr.org/papers/volume15/delgado14a/delgado14a.pdf) does not use it. As said in my review, I think there is no good reason for using it in classification, because F1 does not consider true negatives (which is o.k. in information retrieval, but not in classification). I also do not see a justification for class weighting (but I have not given this much thought).

---

> > ### Author Response · Authors · 2024-11-27
> >
> > - **Additional experiments**: First, let us clear up our setup for RIPPER. We run Ripper with minimum weight in the split of 5% of training samples and default parameters otherwise. Else, in Adult, for example, RIPPER learns over 100 rules, which is completely uninterpretable as you state yourself. By setting 5% as the minimum support, we allow a theoretical maximum of 20 rules, which is double what we allow for the other methods and is more than fair. The code snippet looks like this
> > ```
> > min_weight = 0.05*len(Y_train)
> > Ripper = WekaEstimator(classname="weka.classifiers.rules.JRip",options=["-N",str(min_weight)])
> > ```
> >    We invite you to try the described setting (3 folds, 2 runs, seed 1, check error rate, pruning on).
> > - **Evaluation**: The new manuscript includes RIPPER in the main table. NyRules now stands at rank 2.6, whereas RIPPER achieves 3.5. In terms of accuracy, NyRules is 3.25, while RIPPER is 3.95. In the table we provided on openreview, ties were counted as wins, and we removed a dataset where the optimization of RIPPER failed badly (juvenile) but which was factored into the calculation. We will investigate the reason for this behavior. In the table in the paper, ties in scores are assigned the average rank, i.e., rank 1.5 for two methods with the highest score.
> > - **$F_1$ score**: We firmly disagree with that opinion for three reasons. First, an argument of “other people do not use it” is never a justification to do or not do something. Second, the F1-score is a standard evaluation metric for imbalanced datasets, which is the case for many of the datasets we used, as can be easily seen in Table 3 in the Appendix. Third, questioning F1's validity due to its omission of true negatives misses the metric's core purpose.  The focus of F1 on precision and recall is precisely what makes it superior to evaluate imbalanced datasets, as it unveils if a classifier is simply a constant function and thus does not capture the actual class-conditional distribution. By computing the F1 score for the 0 class, the performance concerning "true negatives" is reflected in the score.
> >
> >    Beyond that, we report the Accuracy in Table 4, where NyRules outperforms the other methods. Hence, if we swap Accuracy and F1, the paper's claims remain the same.
> >
> >
> > - **Difference to RIPPER**: The main differences between RIPPER and NyRules are apparent. RIPPER learns the rules individually, and the list is extended iteratively, whereas we formulate a joint optimization approach using differentiable approximations. RIPPER is a strong competitor, and we do not claim to beat it on every dataset. Still, as the other reviewers agree, our approach is a significant contribution to rule list learning and shows strong empirical results on a variety of benchmarks.

---

> ### Comment · Reviewer_yZdc · 2024-11-28
>
> The default parameter setting for Ripper is weka.classifiers.rules.JRip -F 3 -N 2.0 -O 2 -S 1 (this is what is used in Weka's interactive explorer, and corresponds to what is recommended in Cohen's paper). In this setting, Ripper learns 26 rules on adult, not more than 100, and each of the rules is very simple and interpretable.
>
> ```
> Scheme:       weka.classifiers.rules.JRip -F 3 -N 2.0 -O 2 -S 1
> Relation:     adult
> Instances:    48842
> [...]
> Test mode:    5-fold cross-validation
>
> JRIP rules:
> ===========
>
> (marital-status = Married-civ-spouse) and (education-num >= 14) => class=>50K (2526.0/522.0)
> (marital-status = Married-civ-spouse) and (education-num >= 13) and (occupation = Exec-managerial) and (hoursperweek = 3) => class=>50K (493.0/75.0)
> (marital-status = Married-civ-spouse) and (education-num >= 10) and (education = Bachelors) and (capitalgain = 4) => class=>50K (194.0/1.0)
> (marital-status = Married-civ-spouse) and (education-num >= 10) and (education = Bachelors) and (hoursperweek = 2) => class=>50K (2112.0/714.0)
> (marital-status = Married-civ-spouse) and (education-num >= 13) and (age = 2) and (occupation = Prof-specialty) => class=>50K (169.0/54.0)
> (marital-status = Married-civ-spouse) and (education-num >= 13) and (capitalloss = 3) => class=>50K (49.0/4.0)
> (marital-status = Married-civ-spouse) and (education-num >= 12) and (occupation = Exec-managerial) and (workclass = Private) => class=>50K (130.0/36.0)
> (marital-status = Married-civ-spouse) and (education-num >= 13) and (hoursperweek = 3) => class=>50K (615.0/223.0)
> (marital-status = Married-civ-spouse) and (education-num >= 10) and (occupation = Exec-managerial) and (capitalgain = 4) => class=>50K (46.0/1.0)
> (marital-status = Married-civ-spouse) and (education-num >= 10) and (age = 3) and (fnlwgt >= 206487) and (workclass = Private) => class=>50K (217.0/77.0)
> (marital-status = Married-civ-spouse) and (education-num >= 10) and (occupation = Exec-managerial) and (age = 2) and (fnlwgt >= 113597) => class=>50K (208.0/70.0)
> (marital-status = Married-civ-spouse) and (education-num >= 10) and (age = 3) and (fnlwgt >= 118793) and (occupation = Protective-serv) => class=>50K (47.0/8.0)
> (marital-status = Married-civ-spouse) and (education-num >= 10) and (age = 2) and (fnlwgt >= 176186) and (capitalgain = 2) => class=>50K (46.0/8.0)
> (marital-status = Married-civ-spouse) and (education-num >= 10) and (age = 3) and (fnlwgt >= 162187) and (hoursperweek = 2) => class=>50K (267.0/108.0)
> (marital-status = Married-civ-spouse) and (education-num >= 10) and (age = 2) and (fnlwgt >= 71738) and (hoursperweek = 3) => class=>50K (350.0/152.0)
> (marital-status = Married-civ-spouse) and (education-num >= 9) and (age = 3) and (occupation = Exec-managerial) => class=>50K (323.0/133.0)
> (marital-status = Married-civ-spouse) and (education-num >= 10) and (age = 2) and (fnlwgt >= 190290) and (occupation = Prof-specialty) => class=>50K (63.0/22.0)
> (marital-status = Married-civ-spouse) and (education-num >= 9) and (age = 3) and (hoursperweek = 3) => class=>50K (513.0/254.0)
> (marital-status = Married-civ-spouse) and (education-num >= 10) and (age = 2) and (occupation = Tech-support) => class=>50K (95.0/34.0)
> (marital-status = Married-civ-spouse) and (education-num >= 10) and (capitalgain = 3) => class=>50K (82.0/0.0)
> (marital-status = Married-civ-spouse) and (education-num >= 9) and (age = 2) and (capitalgain = 2) => class=>50K (120.0/42.0)
> (marital-status = Married-civ-spouse) and (education-num >= 10) and (capitalgain = 2) => class=>50K (155.0/60.0)
> (marital-status = Married-civ-spouse) and (education-num >= 9) and (occupation = Exec-managerial) and (age = 4) => class=>50K (265.0/129.0)
> (marital-status = Married-civ-spouse) and (education-num >= 9) and (age = 2) and (occupation = Sales) and (fnlwgt >= 109133) => class=>50K (244.0/120.0)
> (marital-status = Married-civ-spouse) and (education-num >= 9) and (age = 3) and (hoursperweek = 2) and (capitalgain = 2) => class=>50K (54.0/18.0)
>  => class=<=50K (39459.0/5169.0)
>
> Number of Rules : 26
> ```
>
> If I set N to 5% of all the data (-N 2442) I get a single (bad) rule with an extremely big else that captures almost all of the data.
>
> ```
> (marital-status = Married-civ-spouse) and (education-num >= 12) => class=>50K (7359.0/2219.0)
>  => class=<=50K (41483.0/6547.0)
> ```
>
> This is certainly interpretable, but (unlike the previous rule set) it does not yield any substantial insight into the domain.
>
> I don't think this setting with a minimum of 5% support makes sense, it artificially cripples the performance of the algorithm.
> Nor do I think that it makes sense to impose a limit of 10 rules to all possible domains. Some domains will require more rules. Rules are good because (if they are not use in lists) you still have that each individual rule is interpretable.
>
> I think this requires a more precise and deeper analysis than what you currently do in the paper.

---

> > ### Comment · Reviewer_yZdc · 2024-11-28
> >
> > Adding to the above: Here are the results on hepatitis in Rippers default setting:
> >
> > ```
> > === Run information ===
> >
> > Scheme:       weka.classifiers.rules.JRip -F 3 -N 2.0 -O 2 -S 1
> > Relation:     hepatitis
> > Instances:    155
> > Attributes:   20
> >               AGE
> >               SEX
> >               STEROID
> >               ANTIVIRALS
> >               FATIGUE
> >               MALAISE
> >               ANOREXIA
> >               LIVER_BIG
> >               LIVER_FIRM
> >               SPLEEN_PALPABLE
> >               SPIDERS
> >               ASCITES
> >               VARICES
> >               BILIRUBIN
> >               ALK_PHOSPHATE
> >               SGOT
> >               ALBUMIN
> >               PROTIME
> >               HISTOLOGY
> >               Class
> > Test mode:    5-fold cross-validation
> >
> > === Classifier model (full training set) ===
> >
> > JRIP rules:
> > ===========
> >
> > (ALBUMIN <= 2.8) => Class=DIE (13.0/2.0)
> > (PROTIME <= 42) => Class=DIE (15.0/7.0)
> > (SPIDERS = yes) and (BILIRUBIN >= 2) => Class=DIE (11.0/4.0)
> >  => Class=LIVE (116.0/6.0)
> >
> > Number of Rules : 4
> >
> >
> > Time taken to build model: 0.02 seconds
> >
> > === Stratified cross-validation ===
> > === Summary ===
> >
> > Correctly Classified Instances         128               82.5806 %
> > Incorrectly Classified Instances        27               17.4194 %
> > Kappa statistic                          0.4066
> > Mean absolute error                      0.2444
> > Root mean squared error                  0.3663
> > Relative absolute error                 73.9511 %
> > Root relative squared error             90.4509 %
> > Total Number of Instances              155
> >
> > === Detailed Accuracy By Class ===
> >
> >                  TP Rate  FP Rate  Precision  Recall   F-Measure  MCC      ROC Area  PRC Area  Class
> >                  0,438    0,073    0,609      0,438    0,509      0,415    0,698     0,497     DIE
> >                  0,927    0,563    0,864      0,927    0,894      0,415    0,698     0,867     LIVE
> > Weighted Avg.    0,826    0,461    0,811      0,826    0,815      0,415    0,698     0,791
> >
> > === Confusion Matrix ===
> >
> >    a   b   <-- classified as
> >   14  18 |   a = DIE
> >    9 114 |   b = LIVE
> > ```
> >
> > More than 82% from only four little rules, where you get 76%?
> >
> > I can approximate by using your setting, which corresponds to -N 7.75:
> >
> > ```
> > (ALBUMIN <= 2.8) => Class=DIE (13.0/2.0)
> >  => Class=LIVE (142.0/21.0)
> >
> > Number of Rules : 2
> >
> > === Stratified cross-validation ===
> > === Summary ===
> >
> > Correctly Classified Instances         122               78.7097 %
> > Incorrectly Classified Instances        33               21.2903 %
> > ```
> >
> > Yes, yours is simpler, but here it should be very clear that you are overshooting the target.

---

> > > ### Author Response · Authors · 2024-11-28
> > >
> > > ## Hepatitis dataset
> > > We begin with the cited example on the “hepatitis” dataset. We repeat the experiment with 500 different seeds, and analyze the frequency of the 5-fold x-val average accuracy. This experiment can be reproduced with the notebook “hepatitis_dataset_analysis.ipynb” in the supplemental material.
> > > | Bin | Frequency |
> > > | --- | --- |
> > > | 0.6968 - 0.7116 | 3 |
> > > | 0.7116 - 0.7265 | 3 |
> > > | 0.7265 - 0.7413 | 10 |
> > > | 0.7413 - 0.7561 | 60 |
> > > | 0.7561 - 0.7710 | 86 |
> > > | 0.7710 - 0.7858 | 128 |
> > > | 0.7858 - 0.8006 | 142 |
> > > | 0.8006 - 0.8155 | 44 |
> > > | 0.8155 - 0.8303 | 20 |
> > > | 0.8303 - 0.8452 | 4 |
> > > Mean: 0.7796
> > > Our result is obtained with the default seed 1, using our Python-based evaluation pipeline and the sklearn KFold function. Hence, we are most certainly talking about different data folds.  Using the default seed, we obtain an accuracy of 0.76, which is in the center of the observed distribution, whereas the seed you used gives results in the upper range of the distribution. For the other datasets, the differences in results you report, from -0.03 to +0.01,  above the accuracy we report for our cross-validation run. We hope the reviewer agrees that we should compare on expected, rather than best-possible (ie. seed tuned) result. Claims that results obtained with a different setup somehow falsify ours are unfounded and, frankly, unprofessional.
> > > ## Experimental Setup
> > > We continue to welcome an objective discussion about the experimental setup. As explicitly stated in the paper, we compare the performance of rule lists of length {10,15,...}. The only obvious way to control the size of the rule lists returned by Weka RIPPER, i.e., your preferred implementation, is by setting a minimum support. We would be happy to employ a more elegant way if there is one.
> > > ## Reproducibility
> > > We have updated our supplemental material to include the code and bash script for the RIPPER experiment from the rebuttal phase and a requirements file to replicate our Python environment exactly. We fully support the reviewer's right to their personal opinion but firmly reject any insinuation about the integrity of our work.

---

> > > > ### Comment · Reviewer_yZdc · 2024-11-28
> > > >
> > > > I have not insinuated anything. I have just observed that I get different values, for seed 1, and also for seed 2 and seed 3. This can be seen from my original reply. In this reply, I have also reported results that were below the values that you have been reporting (for 2 out of 3 seeds from a different dataset).  Neither of us has reported best-possible (ie. seed tuned) results. You have tried seed 1, I have tried seeds 1, 2, and 3, and reported all three results. Did not try any others, did not try any other datasets than the three I reported.
> > > >
> > > > In any case, reporting the average over 500 different results would be great, you did that now, but not in the paper. Note that I have also criticized that a 5-fold X-val is brittle. 10-fold would certainly have been more stable.
> > > >
> > > > But my main criticism was not that, but that you use a non-sensible setting for Ripper's N parameter that learns rule sets that overgeneralize, and, as the performance of your algorithm is comparable to that, I worry that it suffers from similar problems.
> > > >
> > > > Anyways, I don't think that we will reach a consensus here. I have brought forward my arguments, you have your counter-arguments. I can understand that you are upset (I would be too), this was not my intention.
> > > > I think it is best if we leave that for the area chair to decide. Either way is fine with me, of course.

---

### Official Review · Reviewer_wPvn · 2024-11-03

**Soundness:** 3
**Presentation:** 4
**Contribution:** 3
**Rating:** 5
**Confidence:** 4

**Summary:**

This paper paper proposes a novel end-to-end differentiable model to learn rule lists called NyRules and addresses limitations from existing work.
The method takes into account the discretisation of continuous features into predicates, the learning of conjunctions & rules as well as the ordering of the rules into rule lists.
NyRules performances were validated on both synthetic and real world datasets and compared to existing methods.

**Strengths:**

The paper is clearly written and of great quality.
The solution proposed to solve the vanishing gradients for the chosen computation of the conjunction is well formulated and the efficiency of the relaxation proven with the ablation study.
Supplementary material (with code) is provided for reproducibility.

**Weaknesses:**

Major comments:
- Discretization on the fly has already be done in existing works (for example: _Kusters, Remy, Yusik Kim, Marine Collery, Christian de Sainte Marie, and Shubham Gupta. "Differentiable Rule Induction with Learned Relational Features." In International Workshop on Neural-Symbolic Learning and Reasoning. 2022_). However, the thresholding layer presented in this paper is still very relevant and different from _Kusters et al_ approach for instance.
The paper's contributions need to be put into perspective. Also a comparison between the two different thresholding layers would strengthen the quality of the paper.
- The improvements provided by the rule ordering are not proven. An additional ablation study would be required in order to differentiate the impact of the predicate layer and the active priority. Especially for comparison with RLNet which is also a neural-based approach to learn rule lists.
- The final else (with no predicates) in the rule list is not described nor explained.
- The fact that $l=2$ (binary classification) in the entire paper should be mentioned from start explicitly. It is indirectly mentioned only in 3.4 where the BCE loss is specified, in 5.1 where only datasets for binary classification are used and in future works (6.2). The reader is told this method is applicable for multi-classification with parameter $l$.
- The interpretation of $c_j$ in the rules is not explicit. "if $a_j$ then $c_j$" is then converted in Figure 1 into "if [...] then P(Disease) = 94%" for example. Where is that value coming from ? softmax of $c_j$ ? I believe it needs to be explicit.

Other comments:
- Related work is lacking contributions from different fields. For instance, fuzzy logics with studies like _van Krieken, Emile, Erman Acar, and Frank van Harmelen. "Analyzing differentiable fuzzy logic operators." Artificial Intelligence 302 (2022): 103602._ which analysed different differentiable logical conjunctions.
- The description of the temperature annealing schedule seems to be missing.
- In 3.3 the number of rules $k$ and the number of classes $l$ _are_ (typo?) fixed beforehand.
- Figure 3.c, X1=1 and X2=1 = 0 ?
- References to datasets sources are missing
- Pre-processing of the datasets is not described. How are the datasets continuous feature discretised for other methods ?
- The distribution of the features types in the datasets could be provided to highlight the relevance of the predicate layer of NyRules.

**Questions:**

- Answers / changes following the major comments listed above will address most of the limitations of this current version of the paper.

Additional questions:
- The relevance of the predicate layer is highlighted for the Ring dataset that has exclusively continuous features. Did you observe a drop in performance compared to other methods for datasets with no continuous features ?
- Have you tried other differentiable logical conjunction layers ?

---

> ### Author Response · Authors · 2024-11-18
>
> Dear Reviewer,
> We thank you for your detailed review.
>
> - **Prior Work**: Kusters et al.'s work investigates linear decision boundaries, which generally do not correspond to the desired thresholding functions. Nonetheless, the cited work is relevant, and we will discuss it in the related work section.
> - **Rule Ordering**: That is an excellent idea. We conducted an ablation study in which we re-ran NyRules with a fixed rule order on real-world datasets. We report the F1 Score and the difference compared to the dynamic order below. On average, the performance decreases by 0.04 F1 points. The decrease is correlated with the complexity of the dataset, where more performance is lost on the more complex datasets (Adult, credit card, juvenile).
> | Dataset               |   F1 with Fixed Order |   Difference |
> |:----------------------|----------------------:|-------------:|
> | adult                 |                  0.66 |        -0.14 |
> | android               |                  0.92 |        -0    |
> | compas_two_year_clean |                  0.66 |         0    |
> | covid                 |                  0.64 |         0.02 |
> | credit_card_clean     |                  0.68 |        -0.11 |
> | credit_g              |                  0.64 |        -0.09 |
> | crx                   |                  0.86 |        -0    |
> | diabetes              |                  0.66 |        -0.07 |
> | electricity           |                  0.74 |        -0.01 |
> | fico                  |                  0.68 |        -0.02 |
> | heart                 |                  0.79 |         0.01 |
> | hepatitis             |                  0.76 |        -0.03 |
> | juvenile_clean        |                  0.8  |        -0.07 |
> | magic                 |                  0.77 |        -0.02 |
> | phishing              |                  0.91 |         0    |
> | phoneme               |                  0.59 |        -0.2  |
> | qsar_biodeg           |                  0.79 |        -0.02 |
> | ring                  |                  0.8  |        -0.12 |
> | titanic               |                  0.77 |        -0.01 |
> | tokyo1                |                  0.91 |         0    |
> | Average               |                  0.75 |        -0.04 |
> - **Multi-class classification**: We focus in this paper on differentiably learning the rules and their order, and less so on the consequents. As many rule list methods only support binary classification (SBRL, CORELS, DRS), we evaluated exclusively binary classification. We can extend NyRule to multi-class settings by expanding the dimension of the consequent vector to $l$ where $l$ is the number of classes. We provide preliminary results below for multi-class classification, where we fit rule lists of length n_rules$ = l*5$. Here, NyRules also outperforms the SOTA.
> | Dataset      | NyRules    | RLNet        | CLASSY   | GREEDY   |
> |:-------------|:--------------|:--------------|:---------|:----------|
> | car          | 0.83          | **0.84** | 0.83     | 0.05      |
> | ecoli        | **0.84** | 0.78          | 0.75     | 0.55      |
> | iris         | **0.95** | 0.83          | 0.94     | 0.56      |
> | yeast        | **0.54** | 0.41          | 0.52     | 0.21      |
> | Avg.~Rank    | **1.50** | 2.25          | 2.25     | 4.00      |

---

> ### Author Response · Authors · 2024-11-18
>
> - **Thresholding**: To further examine the effect of thresholding, we also conducted an ablation study of the thresholding component by employing equal-width binning first for all continuous features (3 bins) and re-running NyRules. We report the F1 score and the difference to the thresholding-enabled results below. We observe an average decrease of 0.04 in F1 and up to 0.19 on ring. In general, datasets with mainly continuous features are thus disproportionally affected by static thresholding.
> | Dataset               |   F1 with Fixed Thresholding |   Difference |
> |:----------------------|-----------------------------:|-------------:|
> | adult                 |                         0.75 |        -0.04 |
> | android               |                         0.92 |        -0    |
> | compas_two_year_clean |                         0.59 |        -0.06 |
> | covid                 |                         0.61 |        -0.02 |
> | credit_card_clean     |                         0.79 |        -0    |
> | credit_g              |                         0.66 |        -0.06 |
> | crx                   |                         0.85 |        -0    |
> | diabetes              |                         0.73 |         0    |
> | electricity           |                         0.63 |        -0.12 |
> | fico                  |                         0.65 |        -0.05 |
> | heart                 |                         0.78 |        -0    |
> | hepatitis             |                         0.79 |        -0    |
> | juvenile_clean        |                         0.89 |         0.01 |
> | magic                 |                         0.75 |        -0.04 |
> | phishing              |                         0.91 |         0    |
> | phoneme               |                         0.66 |        -0.13 |
> | qsar_biodeg           |                         0.79 |        -0.02 |
> | ring                  |                         0.73 |        -0.19 |
> | titanic               |                         0.77 |        -0.01 |
> | tokyo1                |                         0.91 |         0    |
> | Average               |                         0.76 |        -0.04 |
> - **Else-case**: the else case is realized through an always-on rule $a(x)=1$ with the lowest priority. We will clarify this in the updated manuscript.
> - **Interpretation of $c_j$**: We model class probabilities using softmax$(c_j)$ and optimize the cross entropy loss. We make this explicit in the updated manuscript.
> About your questions:
> - **Other conjunction layers**: We tried minimum-based logical conjunctions during the early stages of the project but found their performance unsatisfactory. Using a straight-through estimator on the min function may result in arbitrarily bad gradients, while NyRules uses the actual gradient of an approximate logical conjunction.
> - **Continuous vs. discrete datasets** : This is an interesting idea. We take all datasets that have a fraction of at most 20% discrete features and re-run our evaluation. We find that NyRules now has a rank of **1.86** as opposed to 2.30 on all datasets, showcasing that it performs relatively better on continuous datasets.
> - **Temperature schedule**: We use a linear temperature decay during the second half of training for both temperatures. We will include a description of temperature scheduling in the updated Appendix.
> ```
> temp_start = 1.0
> temp_end = 0.1
> temp = temp_start
> step_size = (temp_start - temp_end)/(total_epochs*2)
> If epoch >= total_epochs/2:
> 	temp = temp - step_size
> ```
> - **Pre-processing**: We adopted the pre-processing used by the respective methods. For example, CLASSY employs equal-width binning with 5 cutpoints. We will add this information to the updated manuscript.
> - **Datasets**: We will add each dataset's source and fraction of numeric features to the Appendix.
> We hope this adequately addresses your concerns and questions, and we would be happy to discuss further questions. We will incorporate the proposed changes within a few days into the manuscript and notify you once the updated version is available.
>
> We hope this adequately addresses your concerns and questions, and we would be happy to discuss further questions. We will incorporate the proposed changes within a few days into the manuscript and notify you once the updated version is available.

---

> > ### Author Response · Authors · 2024-11-25
> >
> > Dear Reviewer, the updated manuscript is now available on OpenReview where we have incorporated your valuable feedback. We look forward to your response.

---

> > > ### Comment · Reviewer_wPvn · 2024-11-26
> > >
> > > Dear Authors,
> > > Thank you very much for your answers and updated manuscript. Overall I am satisfied with your answers and changes.
> > > Here are some follow-up comments.
> > > - Prior work: I'm not sure I fully agree with the sentence added in the updated manuscript about Kusters et al paper "focuses on linear decision boundaries which can not be translated into interpretable single feature thresholds" as single feature thresholds learning is a special case and corresponds to the weight $W = I_n$ but that is definitely not the subject and it will not affect my grading.
> > > - Rule Ordering: thank you, this ablation study was required and improves the quality of the paper.
> > > - Multi-class classification: thank you, it also clarifies the paper.
> > > - Thresholding: same.
> > > - Else-case: this clarification does not seem to be in the updated manuscript.
> > > - Interpretation of $c_j$ : Same for that comment, although I did understand, it should be more explicit in the paper.
> > > - Datasets: Although written here I'm afraid I cant find the fraction of numeric features in the Appendix in the updated version.

---

> > > > ### Author Response · Authors · 2024-11-27
> > > >
> > > > We have expanded the table in the Appendix, which now includes the numeric fraction of features per dataset. We have added clarifying sentences regarding the class probabilities and "else" case to the manuscript.
> > > >
> > > > Thank you for helping us improve the manuscript. We kindly ask you to consider updating your score if your concerns have been adequately addressed.

---

### Official Review · Reviewer_nNHw · 2024-11-04

**Soundness:** 3
**Presentation:** 3
**Contribution:** 2
**Rating:** 6
**Confidence:** 4

**Summary:**

The paper introduces Neuro-Symbolic Rule Lists (NyRules), an upgrade to standard rule
lists -- a well known class of white-box classifiers -- that in addition to learning
rules also learn how to discretize continuous input features automatically.  (This
is where the "neuro-symbolic" moniker comes form.)  This feat is achieved by
introducing a continuous relaxation of rule list learning, which is combinatorial
in nature.  NyRules is similar in spirit to existing differentiable rule learning
approaches, except it focuses specifically on rule lists and reuses/improves upon some
key techniques: 1) differentiable thresholding functions from Yang et al.,
used as-is; 2) the differentiable conjunction-of-a-subset operation from Xu et al., improved
to avoid vanishing gradients; 3) linear relaxation + annealing to enable end-to-end learning of
the rule list.  NyRules are evaluated on several data sets against several
competitors.

**Post-rebuttal**: I bumped the score to 6 based on the author's additional experiments and clarifications.

**Strengths:**

**Originality**:  NyRules combines existing ideas and improves on them.  The level of novelty is acceptable.

**Quality**: All technical contributions appear to be good.  The experiments are carried out on several "real-world" data sets as well as on synthetic data, and the results are 5-fold cross-validated (which is refreshing; however: why 5 and not 10+?). The choice of competitors is also sensible.  Performance improvement on "real-world" data is rather robust, although not dramatic in most cases (see also Weaknesses).

**Clarity**: The text is generally well written, and figures/plots are altogether clean and useful.  However, parts of the manuscript feel a bit rushed: I was puzzled by a few equations (which are not always entirely formal) and statements, see below.

**Significance**: NyRules contributes to research on differentiable rule learning, well motivated by explainability requirements.  The contribution is neither niche nor exceedingly significant.

**Weaknesses:**

**Quality**:

- There is a pretty big issue with Table 1, which I wish sorted out:  in many situations, XGBoost performs *much* better than any of the competitors,   yet it is never marked in bold in Table 1.  This is the case also for the rings dataset.  This choice is not explained in the text (I may have missed it!) and risks conveying a twisted perception of performance improvement.

  I understand that boosted trees can be less interpretable than rule lists, and that -- since the paper does not attempt to evaluate interpretability of learned models at all -- this cannot be demonstrated.  But as a result, the results hyperfocus on prediction accuracy, where NyRules does *not* have an edge over XGBoost (to the best of my understanding).

  **Suggestion**: the authors should explicitly explain in the paper why XGBoost results are not bolded, even when they outperform other methods, and clarify that the focus is on comparing interpretable rule-based methods, with XGBoost included as a non-interpretable benchmark. Adding this explanation would provide important context and avoid potential misinterpretation of the results.

- A key issue is that there is not specific study of the impact of the threshold learning step, which is a big selling point of the proposed method.  I am left wondering whether it is indeed important for performance and interpretability.

  **Suggestion**:  the authors could conduct an ablation specifically focused on the threshold learning component by, e.g., comparing NyRules with a version that uses pre-discretized features, to directly demonstrate the impact of learned thresholds on performance and potentially on interpretability.

- The authors equate rule length with interpretability, but this is not exactly exact.

  **Suggestion**: the authors should acknowledge this limitation in the Limitations section discussing that while rule length is used as a proxy for interpretability, a more comprehensive evaluation of interpretability, ideally including user studies, would provide a more accurate assessment.

**Clarity**:

- NeSy is **neuro**-symbolic because the perceptual component -- responsible for encoding low-level inputs such as images into high-level symbolic variables -- is implemented using neural networks.  In this paper the perception component is minimal: it corresponds to the thresholding step.  I honestly feel calling the proposed method "neuro-symbolic" is a stretch, and could be interpreted as disrespectful of what people in NeSy AI aim to do.  I am not going to penalize the paper based on this, but I wanted to make it clear that I did not appreciate this choice.

- Some equations are repeated unnecessarily (the definition of r(.) and eq. 2).

- p2: Last equation: $\alpha \le x_i \le \beta$ is not a function, it is a condition; please add an indicator function.  This also applies to line 184.

- Eq 1: rl(.) is undefined.

- Eq 1: the "s.t." does not make much sense, because this is neither a satisfiability nor an optimization problem.  It should be replaced with an if-and-only-if.  In short, I'd write:
$$
  rl(x) = c_j \qquad \Leftrightarrow \qquad \exists j s.t. (a_j(x) = 1) \land \forall i . p_i > p_j . (a_i(x) = 0).
$$
Since priorities are unique, the condition $i \ne j$ is unnecessary.  Actually, in its current form the equation seems incorrect, as it requires lower-priority conditions not to fire (i \ne j, p_i < p_j), which to the best of my understanding is not needed.  Or did I get this wrong?

- line 188: mangled sentence.

- line 262: what does the argmax_l act on?  The argument is c.

- line 278: same.

- I cannot follow the comment at line 280: what is the point of talking about boosted rule sets at this point?  This is the only place in the paper where they are mentioned.

I am willing to increase my score provided the authors address the main issues I raised.

**Questions:**

- Feel free to discuss any of the main weaknesses I listed above.

- line 204: "In the end, we seek to obtain strict logical rules for use in a rule list."  If I understand correctly, once the temperature is low enough, the learned predicates will already be "almost" discrete.  At this stage, couldn't you just replace them with their hard counterparts, and obtain a hard rule list classifier?  The lower the temperature, the closer the original soft RL classifier and the corresponding hard RL classifier will behave in terms of predictions and loss, meaning there's little to be gained, performance-wise, from using the soft version, and much to be gained (for the few inputs that fall in the affected region of input space), interpretability-wise, by using the hard version.  So why not do that?  Have you evaluated empirically what happens if you follow this route?

- Appendix C: what split did you use for evaluating performance during grid search?

---

> ### Author Response · Authors · 2024-11-18
>
> Dear Reviewer,
> we thank you for your detailed review.
> - **Thresholding**: That is an excellent suggestion. We conducted an ablation study of the thresholding component by employing equal-width binning first for all continuous features (3 bins) and re-running NeuRules. We report the F1 score and the difference to the thresholding-enabled results below. We observe an average decrease of 0.04 in F1 and up to 0.19 on ring. In general, datasets with mainly continuous features are thus disproportionally affected by static thresholding.
> | Dataset               |   F1 with Fixed Thresholding |   Difference |
> |:----------------------|-----------------------------:|-------------:|
> | adult                 |                         0.75 |        -0.04 |
> | android               |                         0.92 |        -0    |
> | compas_two_year_clean |                         0.59 |        -0.06 |
> | covid                 |                         0.61 |        -0.02 |
> | credit_card_clean     |                         0.79 |        -0    |
> | credit_g              |                         0.66 |        -0.06 |
> | crx                   |                         0.85 |        -0    |
> | diabetes              |                         0.73 |         0    |
> | electricity           |                         0.63 |        -0.12 |
> | fico                  |                         0.65 |        -0.05 |
> | heart                 |                         0.78 |        -0    |
> | hepatitis             |                         0.79 |        -0    |
> | juvenile_clean        |                         0.89 |         0.01 |
> | magic                 |                         0.75 |        -0.04 |
> | phishing              |                         0.91 |         0    |
> | phoneme               |                         0.66 |        -0.13 |
> | qsar_biodeg           |                         0.79 |        -0.02 |
> | ring                  |                         0.73 |        -0.19 |
> | titanic               |                         0.77 |        -0.01 |
> | tokyo1                |                         0.91 |         0    |
> | Average               |                         0.76 |        -0.04 |
> - **Marking**: We compare NyRules to other rule list models within its model class and provide XGBoost solely as a reference point for achievable accuracy. We will gray out the column to indicate that we do not consider it a competitor rule list.
> - **Intepretability**: We fully agree that rule length is only one aspect of interpretability. We will update limitations and future work discussing the subjective nature of interpretability and potential ways to evaluate it with human feedback.
> - **Neuro-Symbolic**: We see that NyRules is different from traditional neuro-symbolic methods. Nonetheless, NyRules is a neural method that learns symbolic representations of tabular input and, hence in our opinion neuro-symbolic.
>
> We will adopt your proposed changes to the equations. Thank you for your careful review. We answer your questions below.
> - **Using strict rules**: We translate the model weights into strict rules for evaluation, using the predicate thresholds wherever w_i > 0. We will clarify this.
> - **Grid search**: We perform hyperparameter optimization of 5 separate holdout datasets. Here, we take that configuration, which achieves the average lowest, cross-validated F1 score over all datasets.
> - **Cross validation**: We include datasets with only 100+ samples, where 10 folds could lead to statistically unstable estimates. We decided to use the same number of splits for all datasets and stuck to 5.
>
> We hope this adequately addresses your concerns and questions, and we would be happy to discuss further questions. We will incorporate the proposed changes within a few days into the manuscript and notify you once the updated version is available.

---

> > ### Comment · Reviewer_nNHw · 2024-11-19
> >
> > Thank you for the clarifications and for keeping them short.  I very much appreciate it.
> >
> > - **Thresholding**: We conducted an ablation study [...] we observe an average decrease of 0.04 in F1 and up to 0.19 on ring.
> >
> > So using uniform static thresholds always yields a decrease in performance.  In 8 data sets there is no effect, however.  A few follow-up questions are: 1) Why did you choose 3 bins?  Is this comparable to the number of "bins" learned by NYRules?  2) Do you plan to integrate this result in the main Table? If so, could you please update the PDF accordingly?  3) Is this extra performance worth it, from a computational perspective?  I.e., could you please comment on the relative run-times of NYRules and of the pre-binned variant?
> >
> > - **Marking**: That's what I figured.  I would also add a comment to this effect in the text.
> >
> > - **Intepretability**: "We will update limitations and future work discussing the subjective nature of interpretability" - it is not so much the "subjective" nature of interpretability, it is that objectively sparsity and complexity are different things for human observers.  I would like to know how you plan to update the text and I'd encourage you to update the PDF to illustrate the changes you have in mind.
> >
> > - **Neuro-Symbolic**: I mostly disagree, but as I mentioned I won't penalize the paper based on this.
> >
> > - **Using strict rules**: Thank you.
> >
> > - **Grid search**: *Lowest* cross-validated F1?
> >
> > - **Cross validation**:  Sounds good.
> >
> > As I mentioned, I'd appreciate if you could update the PDF with the changes you mentioned.

---

> > > ### Comment · Reviewer_nNHw · 2024-11-19
> > >
> > > I have one extra remark.
> > >
> > > - **Thresholding**: Could you please check what happens if you use a standard but non-uniform quantization algorithm, e.g., k-means binning (as far as I can see, this is implemented in sklearn)?  The reason I'm asking is that quantization is a well-studied topic, I would expect standard algorithms to behave better than equal-width binning.  Thank you.

---

> > > > ### Author Response · Authors · 2024-11-20
> > > >
> > > > - **Thresholding**:
> > > >    - NyRules learns a lower bound $\alpha$ and an upper bound $\beta$. This corresponds effectively to partitioning the domain into three bins: below, in, and above.
> > > >    We will create a separate subsection for ablation studies, where we examine the respective experiments for thresholding and rule order conducted as part of the rebuttal.
> > > >    - Certain datasets are inherently discrete (e.g., the Android malware dataset, which is fully binary). In such cases, exact thresholding cannot improve performance by definition, which explains why our method does not outperform on every dataset. As per your suggestion, we re-ran the ablation using k-means bins and provide the exact scores in the table below. This does have an impact on the resulting performance, which improves by 0.01 on average but varies more drastically on certain datasets. In general, while fixed binning can sometimes achieve reasonable results, it requires users to tune the discretization manually. On the other hand, the learned discretization by NyRules performs at least as well as the fixed binning and outperforms it on many datasets. We will include these results in the ablation section for completeness.
> > > > | Dataset               |   F1 with K-means Thresholding |   Difference |
> > > > |:----------------------|-----------------------------:|-------------:|
> > > > | adult                 |                         0.66 |        -0.14 |
> > > > | android               |                         0.92 |         0    |
> > > > | compas_two_year_clean |                         0.65 |        -0    |
> > > > | covid                 |                         0.64 |         0.02 |
> > > > | credit_card_clean     |                         0.72 |        -0.07 |
> > > > | credit_g              |                         0.71 |        -0.01 |
> > > > | crx                   |                         0.86 |         0    |
> > > > | diabetes              |                         0.74 |         0    |
> > > > | electricity           |                         0.74 |        -0    |
> > > > | fico                  |                         0.7  |        -0    |
> > > > | heart                 |                         0.79 |         0.01 |
> > > > | hepatitis             |                         0.76 |        -0.02 |
> > > > | juvenile_clean        |                         0.86 |        -0.02 |
> > > > | magic                 |                         0.75 |        -0.04 |
> > > > | phishing              |                         0.91 |         0    |
> > > > | phoneme               |                         0.77 |        -0.02 |
> > > > | qsar_biodeg           |                         0.79 |        -0.02 |
> > > > | ring                  |                         0.72 |        -0.2  |
> > > > | titanic               |                         0.77 |        -0.01 |
> > > > | tokyo1                |                         0.91 |         0    |
> > > > | Average               |                         0.77 |        -0.03 |
> > > >    - Using discretized variables is disadvantageous for NyRules in terms of runtime. This is because we first one-hot encode discrete variables and then use the same thresholding layer on the resulting variables; by discretizing, there are now more variables to process. Therefore, using continuous instead of discretized variables is highly desirable for NyRules.
> > > > - **Marking**: We will clarify the role of XGBoost in the updated manuscript.

---

> > > > > ### Comment · Reviewer_nNHw · 2024-11-20
> > > > >
> > > > > - **Thresholding**: of course, you are right, three bins is a good choice.  Thank you for the clarification.
> > > > >
> > > > > - **Difference column**:  thank you for computing these additional results.  I am not sure I understand the meaning of this column, though:  is this the difference in performance due to using k-means thresholding over NyRules or the other way around?
> > > > >
> > > > > - **Runtime**: makes sense, thanks.
> > > > >
> > > > > - **Interpretability**: I can agree with this, thanks.

---

> > > > > > ### Author Response · Authors · 2024-11-21
> > > > > >
> > > > > > - **Difference column**: It is the performance degradation due to using either uniform or k-means discretization. To be precise: *NyRules-uniform discretization* $-$ *NyRules-Learned discretization* = *Difference*. For example, on the ring dataset, *NyRules-uniform* has an F1 of $0.73$, while *NyRules-Learned discretization*  achieves an F1 of $0.92$, hence the difference is $-0.19$.

---

> ### Author Response · Authors · 2024-11-20
>
> - **Interpretability**: We measure rule length as one aspect of the interpretability of a rule. Empirical evidence shows that shorter rules are more easily comprehensible (Huysmans, 2011), where in that particular experiment, “a drop of approximately 4% to 6% can be observed per size level increment”. Therefore, rule list length and sparsity of the contained rules are regularly used as proxy measures of complexity and, thereby, interpretability (see for example Proenca and van Leeuwen, 2020). They are by no means perfect measures, however. A model that is more easily comprehensible is often not preferred over another more complex one. Fürnkranz et al. present a study where they evaluate the preference and plausibility of rule-based methods, where they observe a preference of study subjects towards more complex rules because they are perceived as more trustworthy.
>
>    In the end, to assess for a particular use case, whether a rule list learned by NyRules is more interpretable and trustworthy than one learned by another method, a user study is required. We do not intend to make any claim that NyRules results in a more interpretable rule list than the current SOTA. What we do instead is to empirically evaluate prediction accuracy, where we find that NyRules comes out on top. We will update the paragraph concerning rule length to make this extra clear and add this discussion to the limitations section.
>  - **Grid-search**: Apologies for the typo. We take the configuration with the highest cross-validated F1.
>
> We will notify you once the updated PDF is available. Thank you again for your constructive review.
>
>
> Huysmans, Johan, et al. (2011). "An empirical evaluation of the comprehensibility of decision table, tree and rule based predictive models." Decision Support Systems 51.1
>
> Proença, Hugo M., and Matthijs van Leeuwen (2020). "Interpretable multiclass classification by MDL-based rule lists." Information Sciences 512
>
> Fürnkranz, Johannes, Tomáš Kliegr, and Heiko Paulheim (2020). "On cognitive preferences and the plausibility of rule-based models." Machine Learning 109.4.

---

> > ### Author Response · Authors · 2024-11-20
> >
> > We have uploaded a revised manuscript. The main changes are as follows:
> > - We have updated the limitations section discussing interpretability and the respective paragraphs in the experiment section.
> > - We now clearly indicate XGBoost as a benchmark method and describe it in text and caption.
> > - We have added a section for ablation studies in the main paper for thresholding and rule order.
> > - We have adopted the changes to notation for enhanced clarity.

---

### Author Response · Authors · 2024-11-24

Based on the valuable feedback of the reviewers, we have incorporated the following changes into the updated manuscript:
- **Ablation studies**: We conducted ablation studies for the thresholding and rule order learning components. We observed that using a static rule order in place of a dynamic one makes NyRules significantly worse for datasets with many samples, while fixed thresholding mainly affects datasets with many continuous features.
- **Multi-class classification**: We provide preliminary results on 4 multi-class classification datasets, finding that NyRules remains the most accurate rule list method.
- **Runtime**: We provide a runtime analysis in the Experiment section.
- **Limitations update**: We have updated limitations with reference to the need for human evaluation of the perceived interpretability of a rule list.
- **Temperature schedule**: We describe our annealing schedule in the Appendix.
- **Clarity**: We have incorporated the suggested changes regarding the notation and other clarity issues.

We thank the reviewers for their constructive feedback and look forward to their response.

---

### Meta-Review · Area_Chair_oxPT · 2024-12-21

**Metareview:**

This paper describes a new approach to learn rule-based models that can handle discretization, rule learning, and rule ordering. The proposed approach -- NyRules --  considers a continuous relaxation of the rule list learning problem that it solves via temperature annealing. This approach can provide a unified way to learn rule-based models without applying pre-processing for discretization . The paper includes experiments on datasets that demonstrate that NyRules consistently outperforms both combinatorial and neuro-symbolic methods across various datasets.

**Recommendation:** Overall, the paper received mixed feedback from four reviewers. On the one hand, most reviewers recognized the novelty of the proposed approach. On the other hand, they raised concerns about computation, the reproducibility of the experiments, and the significance of its practical benefits. In this case, the final evaluation placed this paper at the borderline – and the submission lacked a clear champion.

Having read the reviews, the rebuttal, and the paper, I am recommending rejection at this stage. Overall, I find that there is much to appreciate about the idea in this paper. It proposes a fundamentally different approach to learn rule-based models and shows that they can perform competitively with respect to the state-of-the-art methods. My main concerns in this case pertain to the depth of the evaluation, the strength of the claims, and the significance of the proposed approach. Specifically:

- Limitations / Sandbox: The paper does not include a meaningful study or discussion of the potential limitations of this approach. Just as it is nice to know where the method shines, it would also be helpful to understand where it can fail. I would suggest the authors explore this through synthetic data where they can control the ground truth joint distribution. This would provide a sandbox to evaluate how the relaxation that is used can impact performance and other characteristics under salient distributional assumptions (e.g., linear separability, imbalanced data).

- Demonstration / Unique Functionality: The paper would benefit from a case study that demonstrates new functionality. As it stands, the main takeaway is that the method performs well in terms of F1 score and standard proxies of interpretability (e.g., rule length). And the main qualitative benefits of NyRules over the dozens of alternatives is that we can avoid the need for discretization and reduce training time. A demonstration could provide an opportunity to show why these features may be important, and would provide an opportunity to highlight the improvements in performance and interpretability. One final avenue to improve the paper - related to the ideas above - is to identify new features that are only possible under a differential framework. As an example, a MIP approach provides some unique functionality in that it can allow practitioners to enforce custom constraints on the model and return a certificate of optimality.

**Additional Comments On Reviewer Discussion:**

See above

---

### Decision · Program_Chairs · 2025-01-22

Reject